# Molecular Mechanisms of Canine Osteosarcoma Metastasis

**DOI:** 10.3390/ijms22073639

**Published:** 2021-03-31

**Authors:** Sylwia S. Wilk, Katarzyna A. Zabielska-Koczywąs

**Affiliations:** Department of Small Animal Diseases and Clinic, Institute of Veterinary Medicine, Warsaw University of Life Sciences, Nowoursynowska 159c, 02-776 Warsaw, Poland; sylwia_wilk@sggw.edu.pl

**Keywords:** canine OSA, metastasis, molecular mechanisms, cell lines, in vitro, in vivo, animal models

## Abstract

Osteosarcoma (OSA) represents the most common bone tumor in dogs. The malignancy is highly aggressive, and most of the dogs die due to metastasis, especially to the lungs. The metastatic process is complex and consists of several main steps. Assessment of the molecular mechanisms of metastasis requires in vitro and especially in vivo studies for a full evaluation of the process. The molecular and biological resemblance of canine OSA to its human counterpart enables the utilization of dogs as a spontaneous model of this disease in humans. The aim of the present review article is to summarize the knowledge of genes and proteins, including *p63*, signal transducer and activator of transcription 3 (STAT3), Snail2, ezrin, phosphorylated ezrin-radixin-moesin (p-ERM), hepatocyte growth factor-scatter factor (HGF-SF), epidermal growth factor receptor (EGFR), miR-9, and miR-34a, that are proven, by in vitro and/or in vivo studies, to be potentially involved in the metastatic cascade of canine OSA. The determination of molecular targets of metastatic disease may enhance the development of new therapeutic strategies.

## 1. Introduction

Osteosarcoma (OSA) is the most common bone malignancy that occurs in dogs, accounting for 80–98% of malignant bone tumors [1,2], with an incidence rate estimated to be between at least 5.6 and 13.8/10,000 [3,4,5]. However, it should be pointed out that there is limited data on the statistics on OSA incidence in dogs, as there is no consistent method for reporting cancer in dogs, and the presented estimation is based on data from the 1990s. The origin of neoplasia includes osteoblastic cells producing osteoid or malignant mesenchymal cells. Prevalence is significantly higher in older dogs, with a mean age of 8 years, from large and giant breeds, and in most cases affects the appendicular skeleton (64% of patients) [1,6]. The common locations of this malignancy are the proximal humerus and distal radius (Figure 1A) or the distal femur and proximal tibia (Figure 1B). OSA demonstrates aggressiveness and a propensity to metastasize, most frequently to the lungs [1,2,3] (Figure 1C). Other sites of metastatic disease include the bones and soft tissue of internal organs, such as the liver, spleen, or lymph nodes [2]. It is estimated that at the moment of diagnosis, 80–90% of dogs have micrometastases in their lungs [7]. Despite surgical and chemotherapeutic treatment, less than 20% of dogs survive more than 2 years from diagnosis, probably due to metastatic disease [8].

OSA cells metastasize by hematogenous spread [9]. The process of metastasis is complex and involves several main steps: 1. neo-vascularization within the primary tumor, 2. local invasion and subsequent intravasation, 3. transportation and obstruction at vessels of the secondary site, 4. extravasation and migration, and 5. outgrowth at the secondary site [10] (Figure 2).

Understanding the molecular mechanisms driving metastasis, differences in tumor cells colonizing distant organs from those of primary tumor cells, and recognizing potential molecular targets are important steps in the effective development of anti-metastatic drugs [11]. Both in vitro and in vivo studies are needed to fully evaluate the molecular mechanisms of candidate compounds [12,13]. The in vitro tests can be utilized to assess the invasive phenotype (transwell invasion assay) of canine OSA cells as well as their ability to migrate (wound healing assay (WHA)), which are the main steps of the metastatic cascade. The controlled environment of in vitro studies enables varying expressions of selected factors, which cannot be achieved in vivo [14]. Nevertheless, only in vivo modelling facilitates the mimicking of natural tumor cell behavior and its interactions with the local and systemic environments of a living organism, both crucial for anti-metastatic drug development [15]. Within in vitro cultures, tumor cells lose specific interactions corresponding to tissue formation, which may affect the expression of particular targets [14]. The perfect in vivo model to study preclinical metastasis should produce metastases within a few months, be immunocompetent with the tumor specimen, and be orthotopic [11]. The most widely used in vivo metastatic frameworks are immune-competent mice (engrafted with murine tumors) or immunosuppressed mice (xenografts) [15,16]. Moreover, the primary canine or human OSA tumor implantation into the side of an orthotopic mouse model exemplifies all of the metastatic cascade steps [16]. Other hosts occasionally used in metastatic research include zebrafish, *Drosophila*, and chick embryo chorioallantoic membrane (CAM) [11]. The CAM has been shown to be a reliable model to study neoplastic cell extravasation—an important step of the metastatic cascade [17]. Furthermore, Kim and collaborators demonstrated [17] the potential utility of this model in the development of novel anti-metastatic strategies that specifically target neoplastic cell extravasation.

Spontaneous in vivo models of neoplastic disease are animals that develop the malignancy in a natural manner. Companion animals exhibit resemblances in the pathological characteristics of a specific cancer to its human counterpart, including high incidence rates and response to the therapy, and may be used as a spontaneous in vivo neoplastic model [18]. In vivo tests using a spontaneous canine model of the disease enable evaluation of the potential value of selected targets in pathogenesis and prognosis, especially the evaluation of their expression within the parameters of median survival time (ST) and median disease-free interval (DFI) [7,19,20]. X-ray is a standard technique to detect macrometastases in dogs (Figure 1C). Monitoring macrometastasis in preclinical models is performed with non-invasive imaging methods, such as computed tomography (CT), magnetic resonance imaging (MRI), or bioluminescence [11]. Micrometastases are predicted to be present at the time of the primary tumor diagnosis but are not evident using conventional imaging techniques [11]. Both in preclinical and spontaneous in vivo models, micrometastases can be detected by histological analysis. Substitute or additional methods, such as immunohistochemistry (IHC), flow cytometry of bone marrow aspirates, and molecular assays for tumor-derived DNA in blood samples, have questionable predictive power [11].

The aim of the present review is to summarize the actual knowledge of proteins, genes, and nucleic acids implicated in the process of canine OSA metastasis, such as *p63*, signal transducer and activator of transcription 3 (STAT3), Snail2, ezrin, phosphorylated ezrin-radixin-moesin (p-ERM), hepatocyte growth factor-scatter factor (HGF-SF), epidermal growth factor receptor (EGFR), and miR-9 and miR-34a (Figure 2), as they may act as biomarkers or play an important role in targeted drug delivery. Moreover, a determination of their functions in canine OSA requires investigation due to the biological and molecular resemblance of this tumor to its human counterpart. Canine OSA remains an in vivo model of the human malignancy, utilized in preclinical tests of novel treatment methods [20,21].

## 2. Search Methodology

This review was based on a search of the PubMed database (http://www.ncbi.nlm.nih.gov/pubmed, accessed on 19 February 2021) using the terms “canine” OR “dogs” AND “osteosarcoma” AND “cell line” AND “metastasis”. Out of 62 articles that meet the criteria, only original articles that included information on canine osteosarcoma metastasis at a molecular level were selected. This review is a synthesis of current knowledge in the field and highlights the genes and proteins involved in molecular mechanisms of canine osteosarcoma metastasis that, based on promising results of in vitro and/or in vivo studies, need further investigation. To fully cover a topic, the review also provides the information from other selected canine OSA articles (used in the introduction section) and particular proteins and genes to enhance the knowledge on their biological role.

## 3. Proteins and Genes Potentially Involved in Canine OSA Metastasis—In Vitro and In Vivo Studies

### 3.1. p63

*p63* is a transcription factor that belongs to the *p53* family. *p63* encodes protein isoforms using two promoters: TAp63 and Δp63 [22]. Expression of the Δp63 isoform is often aberrant in several types of human neoplasias, and its upregulation is related to a poor prognosis [22]. Cam et al. [23] demonstrated an overexpression of Δp63 in canine OSA cell lines and primary OSA tumors in comparison to normal canine osteoblasts. In the same study, overexpression of Δp63 enhanced the number of invading D17 cells and improved their ability to migrate in a WHA. It was indicated that D17 cells overexpressing Δp63 exhibit increased phosphorylation of STAT3 and secretion of vascular endothelial growth factor A (VEGF-A), in comparison to control D17 cells. The data suggest that the role of Δp63 in cellular invasion and migration is presumably associated with activating STAT3, interleukin 8 (IL-8), and VEGF-A production [23].

The investigators performed further studies with severe combined immunodeficient (SCID) mice intravenously xenografted with canine OSA cells, which revealed that the expression of ΔNp63 correlates with the metastatic potential of OSA cells and is higher (three-fold more metastases, *p* < 0.05) in the OSA16 cell line when compared to the D17 cell line. In the same study, the correlation between ΔNp63 expression and metastasis was further confirmed by demonstrating an increased lung colonization of D17 cells overexpressing ΔNp63 after IV inoculation in SCID mice, in comparison to control D17 cells. The in vivo experiment data may highlight the involvement of ΔNp63 in the metastatic process of canine OSA. The authors suggest its role in mediating the expression of cytokines, such as IL-8, interleukin-11 (IL-11), and oncostatin M (OSM), but the exact molecular mechanism of ΔNp63 activity in metastasis remains unclear and requires further investigation [23]. Nevertheless, the results of both in vitro and in vivo studies clearly indicate that ΔNp63 is a potential target for novel anti-metastatic therapies for canine OSA.

### 3.2. STAT3

STAT3 is a protein from the family of transcription factors that plays a role in the development, progression, and process of metastasis in some types of neoplasias, including tumors of mesenchymal origin [24]. Activated STAT3 is associated with poor prognosis in some human malignancies, including lymphomas, blood, and solid tumors [25]. Fossey et al. [21] established the activation of STAT3 in numerous canine OSA tumors and canine cell lines. In the same study, the investigators demonstrated that the inhibition of STAT3 with the small-molecule Src inhibitor SU6656 or direct downregulation (by using the small-molecule inhibitor LL3 or siRNA to modulate STAT3 expression) showed a noticeable reduction of mRNAs encoding matrix metalloproteinase 2 (MMP-2) and vascular endothelial growth factor (VEGF), and led to downregulation of the proenzyme and active form of MMP-2 and decreased expression of the VEGF protein. The data suggest that STAT3 activity impacts metastasis and cell invasion through the expression of crucial proteins involved in these processes [21].

### 3.3. Snail2

Various in vitro studies (using the CAM assay and IHC of canine OSA tumors) assessed the role of Snail2, a zinc finger protein and transcription factor from the Snail family, in canine OSA metastasis. Snail2 decreases the expression of E-cadherin, enabling the cells to break cell–cell connections [26]. Immunohistochemical analysis on archived canine OSA showed a significantly higher expression of Snail2 in high-grade canine OSA tumors (n = 11) in comparison to intermediate-grade (n = 5) and low-grade (n = 4) samples [27]. Furthermore, the expression of Snail2 in material derived from appendicular bone tumors was significantly higher than skull-derived OSA [27]. As it is well known that appendicular long bone OSA in dogs metastasizes in 80–90% of cases [1], while skull OSA metastases occur less frequently, according to Sharili et al. [27], this fact may suggest that Snail2 could play a role in the process of cell migration. It should be considered that the study was performed on small numbers of dogs, and archived material was used for IHC, which may limit the results. Downregulation of Snail2 by small interfering RNA demonstrated decreased motility of D17 cells using the scratch assay, suggesting that Snail2 is required for migration [28]. However, overexpression of Snail2 using a Snail2 plasmid with a CMV promoter did not increase the motility of D17 cells. This could be due to the fact that D17 cells metastasize to the lungs and their migration rate cannot be enhanced by increasing Snail2 activity, as they may already be migrating at their highest rate [28]. It was shown that the downregulation of Snail2 can disorganize cytoskeleton architecture without impacting focal adhesions, and D17 cells with increased Snail2 expression exhibited normal cytoskeleton organization and more focal adhesions [27]. This suggests that Snail2 can regulate cell adhesion and cytoskeleton action in OSA, which can be associated with promoting cell migration and metastasis [28]. Further studies on the CAM model demonstrated that the ability to invade the stroma and intravasation of Snail2 knockdown D17 cells was significantly reduced in comparison to control (D17) cells. Results obtained indicated that selectively blocking Snail2 may reduce metastasis in patients with OSA [28]. In various human cancer cells (e.g., lung adenocarcinoma and squamous cell carcinoma), the function of Snail2 in the metastatic processes is considered to be associated with the promotion of MMP expression, but the exact mechanism remains unknown [29,30]. MMPs are proteolytic, calcium-dependent, and zinc-containing enzymes from the metzincin superfamily that degrade and remodel the extracellular matrix [31]. In human medicine, MMP inhibitors were developed as antimetastatic agents. While they inhibited the tissue invasion of neoplastic cells in the metastatic cascade, they failed to show strong efficacy in phase II and III clinical trials [11]. In veterinary medicine, Lana et al. [32] detected significantly higher (*p* < 0.001) MMP-2 (gelatinase A) and MMP-9 (gelatinase B) expression in 30 high-grade OSA samples relative to nearby stromal control tissue. Loukopoulos et al. [33] demonstrated that the synthesis of pro-MMP-9 in canine OSA positively correlates with the histologic grade of the malignancy [34]. In further investigations, Loukopoulos et al. [33] established a noticeable expression of MMP-2 and MMP-9 in three newly derived canine OSA cell lines. Unfortunately, we found no studies of MMP expression in metastatic cell lines nor metastasis in canine patients with OSA despite clear evidence of correlation between MMP-2 and MMP-9 with tumor malignancy. As a result, it cannot be clearly stated whether MMPs play an important role in the metastatic cascade of canine OSA, and further molecular and in vivo studies are needed.

### 3.4. Vimentin

Vimentin is a structural protein expressed in cells of mesenchymal origin and accounts for the main constituent of their cytoskeleton. Physiologically, it is involved in sustaining structural integrity, flexibility, the shape of cells, as well as the processes of migration and adhesion [35,36]. In neoplasias of epithelial origin, vimentin serves as a biomarker of the epithelial-to-mesenchymal transition (EMT) [37]. During the EMT process, epithelial cells acquire the ability to migrate and invade, which is crucial to initiate the metastatic cascade [38,39]. Despite OSA’s mesenchymal origin, human OSA cells are capable of undergoing EMT, leading to increased motility and invasiveness [40]. Overexpression of vimentin in canine OSA is suspected to be associated with increased tumor cell motility, invasiveness, and a more aggressive phenotype [41]. Roy et al. [35] noticed a higher (over 3-fold) expression of vimentin in the canine OSA metastatic cell line HMPOS versus the non-metastatic cell line POS. A similar association was observed in their human counterparts, where expression of vimentin was higher in the metastatic 143B cell line versus the non-metastatic HOS cell line, although the increase was only 1.2-fold higher [35].

### 3.5. Growth Factors and Their Specific Receptors

The implication of growth factors in promoting the invasive phenotype of neoplastic cells has been described in various human and canine malignancies, including OSA [42,43].

#### 3.5.1. Vascular Endothelial Growth Factor (VEGF)

Plasma VEGF is a homodimer signal protein and a member of the platelet-derived growth factor family. Physiologically, VEGF is a mitogen for cells of endothelial origin and promotes angiogenesis [44,45]. Expression of this protein in primary tumors is associated with poor prognosis and high rates of metastasis in some types of human malignancies, including carcinomas of the gastrointestinal tract, hepatocellular carcinoma, and OSA [46,47,48]. In human OSA tumors, the VEGFA level was positively correlated with the presence of lung metastasis [48]. In vitro studies performed by Cam et al. [23] determined that ΔNp63 contributes to the production of VEGF in canine OSA through effects on STAT3 and IL-8. Depletion of ΔNp63 in canine OSA cells resulted in a decreased level of VEGF and a noticeable reduction of endothelial tube formation in transwell endothelial tube formation assays. These findings suggest that VEGF may be involved in the metastatic cascade of canine OSA by promoting the process of angiogenesis [23].

#### 3.5.2. Insulin Growth Factor 1 (IGF-1) and Its Receptor

IGF-1, which plays a part in promoting the metabolic and growth effects of cells, is a polypeptide structurally homological to proinsulin [49]. Regulating the process of osteoblast proliferation, IGFs are one of the most significant growth factors in human bone tissue. In culture, both human osteoblasts and OSA cells have receptors for IGF-1 and are able to proliferate in response to IGF-1 [42]. In veterinary medicine, treating cells with IGF-1 was shown to increase invasiveness in just one of three osteosarcoma cell lines (with high insulin-like growth factor receptor 1 (IGF1-R) expression) [42]. This suggests a cell line-dependent role of IGF-1 in canine OSA invasive phenotypes. Furthermore, the investigators performed studies on immunodeficient mice indicating that aggressive in vivo behavior (tumorigenesis and occurrence of metastasis) of canine osteosarcoma cells positively correlates with the expression of IGF1-R [42]. However, the study was conducted on only three mice per group, which reduced the power of the study and may be unrepresentative of the population. Data regarding IGF1-R in canine OSA is limited to one study where immunostaining was performed and showed a shorter survival time in canine appendicular OSA patients with high IGF1-R immunoexpression compared to those with low IGF1-R [19]. Unfortunately, this study lacked further information on the role of IGF-1 and IGF1-R in the metastatic cascade.

#### 3.5.3. Transforming Growth Factor Beta (TGFβ)

TGFβ is a secreted cytokine that exists in isoforms: TGFβ 1, 2, and 3 [50]. All of the isoforms regulate the differentiation of mesenchymal stem cells, which affects the skeletal development of embryos. Postnatally, TGFβ plays a part in bone homeostasis, including differentiation, proliferation, the survival of osteoblasts, as well as the motility of their precursors [51]. Activity of TGFβ is mediated by two different pathways: the canonical (Smad dependent) and non-canonical (Smad independent) [50]. In human and canine OSA, the factor is implicated in the growth, migration, and invasion of tumor cells [52,53]. The expression of transforming growth factor beta receptors (TGFβRI and TGFβRII) in both normal osteoblasts and five malignant canine OSA cell lines indicated that the TGFβ signaling pathway is not characteristic of the malignant phenotype but is active in reparative osteoblasts [53]. The authors used a WHA to assess an alteration in canine OSA cell motility after blocking a Smad-dependent TGFβ-mediated pathway (through treatment with LY2109761, a small-molecule inhibitor of TGFβRI and TGFβRII). Decreased migration was evaluated as cell line dependent, with the greatest reduction in the D17 cell line, moderate in the HMPOS cell line, and minimally altered in the Abrams cell line. Complete blocking of both canonical and non-canonical signaling did not result in diametrical inhibition of cell motility [53].

#### 3.5.4. HGF-SF and Its Receptor (Met Receptor)

HGF-SF is a heterodimer that consists of two subunits: α and β [54]. The factor is produced by cells of mesodermal origin, and for the most part impacts the biological functions of epithelial and endothelial cells [55]. HGF-SF is a mitogen for certain cell types, such as hepatocytes, keratinocytes, or kidney epithelium [54]; it also stimulates cell motility [56]. The cell surface receptor for this growth factor is the c-Met tyrosine kinase receptor, encoded by *c-MET* [54]. Normal osteoblasts and osteoclasts express c-Met, but secretion of HGF-SF occurs only in osteoclasts. The factor is an inducer of proliferation in both osteoblasts and osteoclasts, and in osteoclasts, it affects cell motility [57]. Upregulation of c-Met was described in human and canine OSA [58,59,60]. The expression of c-Met can be associated with the metastatic phenotype of human OSA [61]. High expression of c-Met mRNA was demonstrated with Northern Blot and real-time PCR (RT-PCR) analysis of both canine OSA cell lines [57] and five of seven canine OSA biopsies [58]. Met-inhibiting molecules (met-specific interfering RNA and small-molecule inhibitor of Met catalytic activity PHA-665752) reduced the invasiveness of canine OSA cells [60]. The data suggest a contribution of HGF and the Met receptor on the metastatic phenotype of canine osteosarcoma [58,60].

#### 3.5.5. EGFR (ErbB1, HER1)

EGFR belongs to the family of ErbB receptor proteins, containing four members: ErbB1, ErbB2, ErbB3, and ErbB4 [43]. This transmembrane receptor consists of two domains: extracellular, which is the ligand binding domain, and cytoplasmic, which encodes the EGF-regulated tyrosine kinase [61]. The major role of tyrosine kinase is to regulate cell proliferation [62]. In tumor development, deregulation of EGFR signaling results in the enhancement of cell proliferation, motility, and angiogenesis, as well as a decrease in apoptosis [63]. In human OSA, high expression of EGFR is associated with the occurrence of distant metastases and poor outcomes [64]. A noticeably higher expression of EGFR was determined in primary and metastatic canine OSA tumors compared to normal bone tissue specimens, as well as in several canine OSA cell lines [43]. Furthermore, the expression of EGFR was significantly higher in canine primary OSA metastases to the lungs, in comparison to extrapulmonary OSA metastases. The results indicate a potential role of EGFR in promoting an aggressive phenotype in canine OSA, although a potential molecular mechanism remains undescribed [43].

### 3.6. Integrins

Integrins are a family of glycoprotein receptors, heterodimers consisting of two subunits: alpha and beta. Their biological function is to regulate many cellular processes, including cell adhesion, migration, and metastasis [65]. Integrins play a part in the invasiveness of tumor cells by activating MMPs and consequently inducing a disintegration of the extracellular matrix (ECM) [66]. In human medicine, high expression of β1 integrin was shown for OSA metastatic cell lines; it was not identified in two canine OSA cell lines (non-metastatic POS and artificially generated metastatic-HMPOS [35]). However, a strong limitation of the study was that it was performed only on two OSA cell lines: POS, derived from a primary spontaneous canine femoral OSA, and the HMPOS, which was derived from POS after serial implantation into the lungs of nude mice [67]. In veterinary medicine, RT-PCR analyses revealed that β4 integrin seems to be associated with an invasive phenotype of canine OSA cell lines. Additionally, inducing β4 integrin genes in non-invasive canine OSA cells led to their acquirement of an invasive phenotype [68]. However, further in vitro and in vivo studies should be performed to confirm the role of β4 integrins in the metastatic cascade.

### 3.7. CD147

CD147 (basigin; EMMPRIN-extracellular MMP inducer; cluster of differentiation CD147) is a plasma membrane protein of the immunoglobulin superfamily, with variable expression in a number of cell types, such as leukocytes, hematopoietic, endothelial, and epithelial cells [69]. Physiologically, it is involved in the regulation of biological processes, such as spermatogenesis [70], lymphocyte responsiveness [71], and MMP synthesis [69]. The expression of CD147 is also described in tumors, including human and canine OSA [35,72]. Stimulating the production of MMPs in neoplasias, CD147 implicates tumor invasiveness and the promotion of metastasis [72,73]. Roy et al. [35] established a 2.12-fold change in CD147 expression in HMPOS (artificially derived metastatic cell line) versus POS (non-metastatic cell line) and further confirmed it by western blot (WB), flow cytometry, and IHC of cell pellets. Similar results were obtained for canine patient samples (although examining only two distinct sets—with five sections of each—from paired primary tumor and lung metastasis samples), although the intensity of CD147 in IHC was not as intense as that in cell lines. The obtained results suggest that CD147 may potentially be considered as a metastatic biomarker and therapeutic target in canine OSA [35]. However, further studies of more cell lines (especially primary metastatic cell lines) and many more canine samples from both primary and metastatic OSA are obligatory to confirm this hypothesis.

### 3.8. Collagen

Collagen is the major structural protein component of the extracellular matrix of connective tissues [74]. It is suggested that enhanced collagen density may initiate a promotion of tumorigenesis, invasion, and metastasis in human breast carcinoma [75]. In human OSA, intercommunication between tumor cells and type I collagen mediates MMP-2 synthesis and activation [76]. Forty-six percent higher expression of collagen was demonstrated in the metastatic canine OSA cell line HMPOS versus the non-metastatic cell line POS [35]. The potential role of collagen in cell migration was further confirmed by experiments treating canine OSA cells with an inhibitor of P4AH1, a protein involved in collagen biosynthesis [77]. Inhibitor-treated HMPOS cells showed a higher dependency of cell migration on collagen synthesis under normal and hypoxic conditions, in comparison with the POS cell line. Song and collaborators also established that hypoxic conditions lead to upregulation of four proteins (P4HA1, PLOD1, PLOD2, and LOX) that are involved in collagen synthesis and a remodeling pathway in metastatic HMPOS [77]. Nevertheless, the strong limitation of both of the presented studies is that they were performed only on one metastatic OSA cell line (HMPOS).

### 3.9. Ezrin and p-ERM

Ezrin is a membrane-cytoskeleton linker that belongs to the ERM (ezrin/radixin/moesin) family. Ezrin plays a part in various biological processes, including cell adhesion and motility. The role of this protein is to link the plasma membrane to cytoplasmatic actin filaments [78]. This molecular connection enables an interaction between cells and their microenvironment and is proven necessary for metastasis [79]. Expression of ezrin has been identified in many human malignancies, with most significant aberrations in mesenchymal neoplasias [80]. Overexpression of this protein is associated with poor outcomes and a high propensity to metastasize. Jackson et al. [68] observed a correlation between higher ezrin mRNA expression and an invasive phenotype and found that expressing ezrin in non-invasive canine OSA cells enhanced their invasive potential. Ezrin is located in an inactive state in the cytoplasm. Phosphorylation of threonine and tyrosine at the C-terminus of the protein leads to conformational activation of its structure and enables a linkage between the C-terminus and actin cytoskeleton, as well as a connection between the N-terminus and cell membrane or its proteins [81].

Ezrin phosphorylation is regulated in a dynamic manner by protein kinase C (PKC) during metastatic progression in both human and canine OSA. To determine the involvement of PKC in the phosphorylation of ezrin in canine OSA, the WHA was performed on four patient-derived ezrin-expressing canine OSA cell lines treated with specific small-molecule inhibitor of PKC [82]. Inhibition of PKC resulted in a noticeable decrease in cellular migration. The data suggest that PKC directed ezrin phosphorylation and the consequential migration of canine OSA cells [82]. An increased expression of ezrin and p-ERM was also demonstrated in vivo in HMPOS tumor tissue that metastasize to the lungs of mice 1 week after IT transplantation [83]. The expression of p-ERM decreased after 2 and 4 weeks. A non-metastatic cell line showed low or no expression of ezrin and p-ERM [83]. These results indicate a potential association between the expression of ezrin and p-ERM, the ability of canine OSA to metastasize, and an involvement of ezrin phosphorylation in early steps of the process [83]. In canine patients, a positive expression of ezrin and p-ERM was shown in 83% of spontaneous primary canine OSA samples [80]. Unfortunately, the study was performed only on primary OSA, not metastatic disease [81]. Dogs entered the study with no evidence of pulmonary metastasis on thoracic radiographs and no clinical evidence of metastasis to the other sides. However, the likely presence of micrometastasis can be assumed. The observed increase in ezrin and p-ERM in the primary tumor, combined with results from the study of Jaroensong et al. demonstrating [83] increased ezrin and p-ERM in the first week following IT tumor transplantation, may further support the idea that ezrin and p-ERM are involved early in the metastatic process. Early involvement may allow primary tumor cells to intravasate and transport to the lungs, after which p-ERM levels decrease as the metastatic tumor is established. Nevertheless, further investigations are required to evaluate the exact role of ezrin and p-ERM in the molecular mechanisms of canine OSA lung metastasis.

### 3.10. Yes-Associated Protein (YAP) and Transcriptional Coactivator with PDZ-Binding Motif (TAZ)

YAP and its paralog TAZ (also called WW domain-containing transcription regulator protein 1, WWTR1) are transcriptional coactivators that play a part in the transcription of genes implicated in cell proliferation and the suppression of apoptosis [84]. These proteins are expressed in almost every human solid tumor [85]. In the metastatic processes, YAP and TAZ facilitate cell migration, enhance cell survival in circulation and at secondary sites, and promote metabolic adaptation to the new environment [86]. TAZ and YAP mediate TGFβ-induced carcinoma cell invasion and motility [87]. In metastatic canine OSA cell lines, TAZ depletion resulted in a significant decline in migratory ability, as opposed to non-metastatic primary tumor-derived cell lines [87]. Additionally, simultaneous TGFβ treatment lead to a noticeable reduction of migration only in the metastatic cell line, not in those of primary tumor origin [88]. The data suggest that the modulatory effect of YAP, TAZ, and TAZ-mediated TGFβ signaling on migration is cell line dependent. However, the impact of TAZ on the migration of metastatic-derived canine OSA cell lines is more distinct than the effect of YAP [88].

### 3.11. The Tropomyosin-Related 00 A (TrkA)

TrkA is a receptor that mediates differentiation, mitogenesis, and survival of neurons, through binding of neurothropin growth factors [89]. It is described that inhibition of TrKA signaling results in a reduction of proliferation and enhancement of apoptosis in murine and human immortalized osteoblast cell lines. Furthermore, the TrkA signaling pathway is believed to be involved in the proliferation and survival of canine OSA cells in both local and metastatic microenvironments [90]. Fan and collaborators [90] determined the positive membranous immunostaining of TrKA in most OSA cells in 66% (10/15) of primary bone lesions and in 75% (9/12) of spontaneously occurring canine OSA metastases, suggesting that TrKA serves as a potential therapeutic target of both primary and metastatic canine OSA [90].

### 3.12. C-X-C Chemokine Receptor Type 4 (CXCR-4)

CXCR-4 is associated with tissue-specific metastases in human malignancies, due to the high concentration of its endogenous ligand-CXCL12, including metastases to the lungs, bones, lymph nodes, and the liver [91]. Fan and collaborators [92] discovered high expression of CXCR-4 in canine osteosarcoma cell lines and in 2 of 10 canine OSA pulmonary metastases. In the same study, the authors assessed that ligation of CXCR4 with exogenous CXCL12 induced directional migration of canine OSA cells [92]. Byrum and collaborators [93] observed the impairment of directional migration of canine OSA cells after reduction of CXCR4 following zoledronate treatment [93]. This result is consistent with the previously described research [92] and may indicate that the CXCR-4 contribution in canine OSA directs migration as a part of the metastatic process [92,93]. Further studies of both in vitro and in vivo on more samples are needed to confirm this hypothesis.

### 3.13. microRNA (miR-9 and miR-34a)

Micro RNAs (miRNAs) are small single-stranded nucleic acids that regulate the expression of the protein-coding genes following transcription. Affecting genes in many molecular pathways, miRNAs regulate physiological processes significant to cell homeostasis [94]. It is well described that alterations of miRNA expression commonly occur in human neoplasias, and they can target genes involved in the genesis, progression, and metastasis of tumors [95]. In both human and canine OSA, aberrant expression of miRNA is associated with poor prognosis due to a higher risk of metastasis and decreased response to chemotherapy [96].

Fenger et al. [96] observed an upregulation of miR-9 in canine OSA cell lines and in primary OSA tumors, in comparison to normal canine osteoblasts. Canine OSA cells overexpressing miR-9 exhibited enhanced invasiveness and migration in comparison to control cells. Inhibition of miR-9 in canine OSA cells resulted in their reduced migration and invasion [97]. These findings support the presumption that miR-9 plays a role in the promotion of migration and invasion in canine OSA cells. One protein regulated by miR-9 that may promote the metastatic phenotype is gelsolin. Diminishment of gelsolin in several types of neoplastic cells resulted in a reduction of their motility [97].

Lopez et al. [98] reported decreased expression of miR-34a in canine OSA tumors and OSA cell lines in comparison to normal canine osteoblasts. OSA cells transduced with pre-miR-34a lentiviral vector show reduced invasion, cell motility, and scattering [98]. Lopez et al. [98] identified Kruppel-like factor 4 (KLF4), Semaphorin 3E (SEMA3E), and VEGFA in transduced and non-transduced OSA cells. These genes are associated with cell migration, and their transcripts are downregulated in cells with mi-R34a overexpression [98]. KLF4, SEMA3E, and VEGFA are presumed as miR-34a target genes, and the results imply that miR-34a may play a role in canine OSA metastasis by regulating their expression [98]. In further studies, Lopez and collaborators [97] showed a 50% reduction of miR-34a expression in metastatic lesions in comparison to primary tumors. However, the limited number of the analyzed samples did not enable the establishment of strong conclusions about miR-34a expression and its role in metastases of canine OSA [98].

### 3.14. Annexins

Annexins are cellular proteins possessing the ability to dependently bind negatively charged phospholipids on calcium [99]. Annexins consist of two domains located on COOH and NH2 terminals, which are responsible for binding cell membrane phospholipids and cytoplasmic proteins, respectively. The major role of annexins is to provide a membrane scaffold for cells [100]. In humans, studies show a negative correlation between the expression of annexin 1 and the progression of breast carcinoma [101]. In canine OSA, Roy et al. [35] determined a significant decrease in annexin 1 production in the highly metastatic cell line HMPOS in comparison to the non-metastatic cell line POS. Conversely, the expression of annexin 2 is noticeably higher in the HMPOS cell line than in the POS cell line [35].

### 3.15. Tissue Factor (TF)

Tissue factor (TF) (also called thromboplastin or coagulation factor 3) is a transmembrane protein that initiates the conversion of prothrombin to thrombin in the coagulation cascade [102]. Through procoagulant and signaling activity, TF contributes to growth processes, angiogenesis, and metastasis in human malignancies, including colorectal carcinoma, breast carcinoma, melanocarcinoma, and glioblastoma [103]. Enhanced expression of this protein is correlated with shorter ST in human lung carcinoma [104]. In dogs, expression of TF is significantly higher in neoplasias of epithelial origin, such as pulmonary adenocarcinoma and mammary gland tumors, in comparison with mesenchymal malignancies like canine fibrosarcoma [105]. Stokol et al. [105] indicated a cell line-dependent expression of this protein in canine OSA. The authors noticed zero or minimal expression of TF in HMPOS and D17 cell lines, while in the OS2.4 cell line, the amount of TF resembled the expression in fibrosarcoma cells [105]. Differences in expression among particular cell lines can be associated with in vivo tumor behavior, a point Stokol and collaborators imply is indispensable for further investigations [105].

## 4. Conclusions

In Vitro and in vivo studies concerning canine OSA metastasis provide information on the specific biology and behavior of malignancies. Moreover, establishment of the specific factors involved in this process enables a broad understanding of its molecular mechanisms, effects on specific steps of the metastatic cascade, and may assist in the revelation of new therapeutic strategies in metastatic disease. Several proteins were examined for their role in the metastatic cascade of canine OSA (Table 1) in vitro or/and in vivo, using either induced animal models or a spontaneous canine model of the disease. The present review summarizes the results of up-to-date knowledge on genes and proteins potentially involved in canine OSA metastasis and points to further directions investigators should consider following to truly understand the molecular pathways involved in canine OSA metastasis. The strongest evidence evaluating both in vitro and in vivo studies presented for proteins involved a particular step of the metastasis cascade: EGFR for neoplastic cell migration; HGF-SF and Met-receptor for invasion; and ΔNp63, ezrin, p-ERM, Snail2, miR-9, and miR-34a for both migration and invasion. Further validation of in vitro results (on several metastatic cell lines) with in vivo methods (with a large enough sample size) is needed to portray all steps of the canine OSA metastatic process and to further develop novel therapeutic strategies.

## Figures and Tables

**Figure 1 ijms-22-03639-f001:**
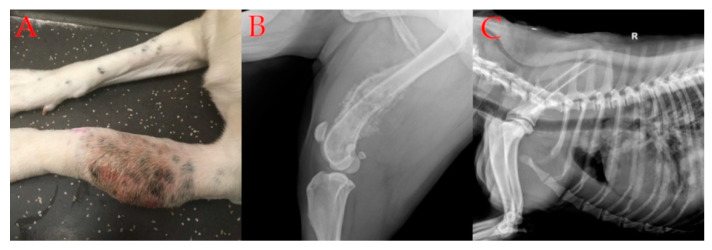
(**A**): Clinical appearance of an 11-month mixed breed dog with osteosarcoma; (**B**): X-ray of the primary canine osteosarcoma localized in the distal femur; (**C**): X-ray of lung metastases in canine osteosarcoma.

**Figure 2 ijms-22-03639-f002:**
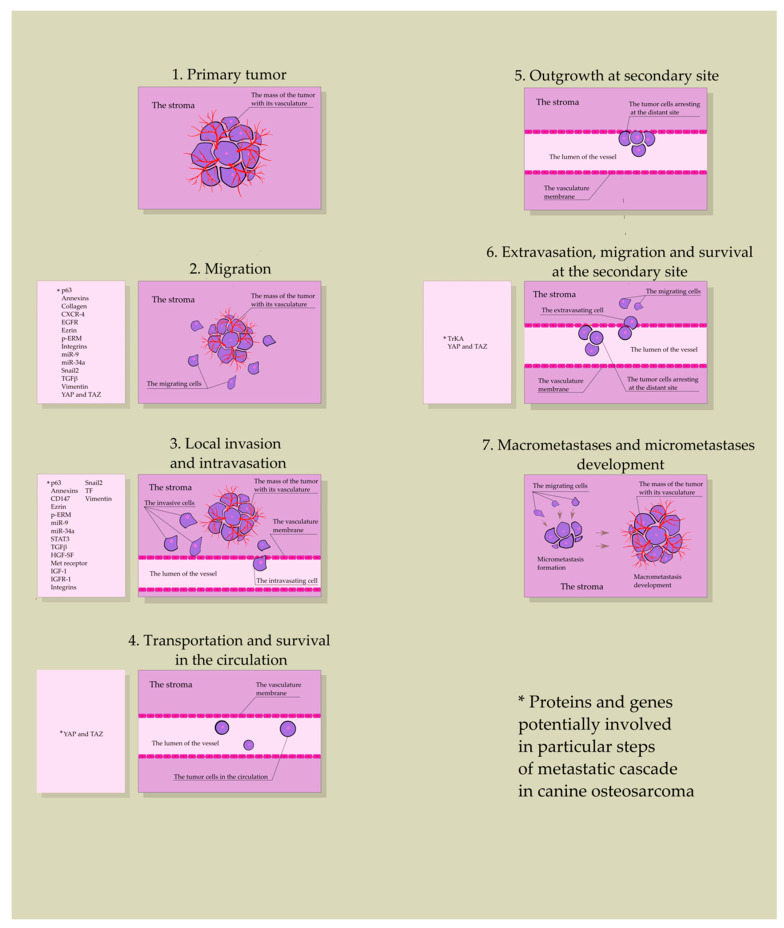
Steps of the metastatic cascade in canine osteosarcoma, including genes and proteins potentially involved in each step.

**Table 1 ijms-22-03639-t001:** Proteins and genes potentially involved in canine OSA metastasis evaluated with particular in vitro and/or in vivo methods.

Protein/Gene	In Vitro		In Vivo		Step(s) of the Metastatic Cascade
	Method	Cell Lines	Method	Samples	
*p63*	qRT-PCR; WB; TIA; quantitative migration assay; WHA [23]	Abrams, OSA8, OSA16, D17 [23]	necropsy and light microscopy [23]	SCID mice [23]	invasion; migration [23]
Annexins	peptide fingerprinting [35]	HMPOS, POS [35]	-	-	Invasion; migration [106,107]
CD147	peptide fingerprinting; WB; flow cytometry; confocal microscopy; IHC [35]	HMPOS, POS [35]	IHC [35]	spontaneously occurring canine osteosarcoma [35]	Invasion [72,73]
Collagen	peptide fingerprinting [35]; WHA [35,77]; nano-Lc-MS/MS analysis, parallel reaction monitoring (pRM) MS, WB [77]	HMPOS, POS [35,77]	-	-	Migration [77]
CXCR-4	directional migration assay [92,93] RT-PCR; WB; flow cytometry; IHC [92]	POS, HMPOS, COS31, Buck, D17 [92] K003 [93]	IHC [92]	spontaneously occurring canine osteosarcoma [92]	Migration [92,93]
EGFR	WB; IHC; qRT-PCR [43]	COS31, HMPOS, POS, D17, KOS-001, KOS-002, KOS-003, KOS-004 [43]	tissue microarray; IHC; qRT-PCR [43]	spontaneously occurring canine osteosarcoma [43]	Migration [63]
Ezrin p-ERM	Matrigel IA qRT-PCR; WHA [82]	MC-KOSA, SK-KOSA, BW-KOSA [68] KOS-001, KOS-002, KOS-003, KOS-004 [82]	IHC; necropsy; [82,83]	SCID mice [82] BALB/c nude mice [83]	invasion; migration [68]
HGF-SF Met receptor	Matrigel IA [58,60]; Northern blot; qRT-PCR [58];	D17 [58,60], D22 [60] Abrams, Grey [58]	Northern blot analysis; IHC [59]	spontaneously occurring canine osteosarcoma [59]	Invasion [58,60]
IGF-1 IGF1-R	Matrigel IA; WB; qRT-PCR [42,68], Northern blot [42]	Abrams, Grey, D17 [42], SK-KOSA, MC-KOSA, BW-KOSA [68]	Necropsy; karyotypic analysis; [42]	athymic nude mice [42]	Invasion [42,68]
Integrins	peptide fingerprinting [35]; RT-PCR [68]	HMPOS, POS [35], MC-KOSA, SK-KOSA, BW-KOSA [68]	-	-	invasion; migration [66,68]
miR-9 miR-34a	Matrigel IA; WHA; qRT-PCR; Nano-LC/MS/MS; [97,98], WB [97]	OSA8, OSA16 [97,98], OSA2, OSA40, OSA50, Abrams, D17 [98]	qRT-PCR [98]	spontaneously occurring canine osteosarcoma [97]	invasion, migration [97,98]
Snail2	WHA [28]; qRT-PCR [108]	D17 [28], OSCA-8 [108]	fluorescent microscopy [28]	the CAM model; [28]	invasion, migration [28,109], intravasation [28]
STAT3	WB; RT-PCR; gel zymography [21]	OSA8, OSA11M, OSA16, OSA29, OSA32, D17 [21]	-	-	Invasion [21]
TF	RT-PCR; flow cytometry; immunofluorescent microscopy [105]	HMPOS, D17, OS2.4 [105]	-	-	Invasion [110]
TGFβ	WHA [53]	HMPOS, Abrams, D17 [53]	-	-	Invasion, migration [52,53]
TrKA	-	-	IHC [90]	spontaneously occurring canine osteosarcoma [90]	survival of the tumor cells in metastatic tumor microenvironment [111]
Vimentin	peptide fingerprinting; confocal microscopy; WB [35]	HMPOS, POS [35]	-	-	invasion, migration [41]
YAP and TAZ	migration transwell assay [88]	D17, OVC-cOSA31, OVC-cOSA-75, OVC-cOSA-78 [88]	-	-	Migration [85,88], survival in circulation and at the secondary sites [86]

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
