# Peer review of "Molecular Mechanisms of Canine Osteosarcoma Metastasis"

_ijms, 2021, doi:10.3390/ijms22073639_

Round 1

Reviewer 1 Report

The authors have reviewed some of the molecular mechanisms of OS metastasis. There are numerous problems with the English and its usage.

  1. Table 1 is almost unreadable. May be lines between the each of Proteins or genes would help. If there is a gene like p63 it must be in italics and that would mean that you do not have to stipulate gene.
  2. Figure 1- I am not sure why this Figure is here. Upper panel- If you are going to mention the non-stochastic pathogenesis of metastases then tie it in with your list of factors/ genes. Which factor/gene relate to each of these steps. Also quote on of the Fidler papers -may be his 2003 Nature Reviews Cancer review examining ‘seed and soil’ hypothesis and then tie into Table 1. Lower panel- what is it saying? If it does not make a point then remove it. Location of organs is meaningless.
  3. Need to check the spacing as words run together like in line 106.
  4. First line of the abstract- “canines” has to be replaced by “dogs” This is also found in other places in this manuscript. Please correct this.
  5. “Mice” must replace “murines” in lines 57 and 58
  6. Many of the sentences in the manuscript are over 30 words long and that results in the meaning being lost. This has happened throughout the manuscript and this is a basic mistake.
  7. Also there is inconsistent use of italics in “in vivo” and ”in vitro”. Just choose one and stick to it.
  8. Line 28 remove “a high”. Aggressiveness is enough and then it is “a propensity”
  9. Line 31 it is “diagnosis”
  10. Line 33 insert “probably” before “due to” as there all the dogs were not autopsied.
  11. Line 61 the language is too extravagant. Remove “definitively”.
  12. Lines 67- remove sentence starting “in the European..” Rewrite next sentence.
  13. Line 94- put in an original reference for p63!
  14. Line 104- rewrite the sentence- really awkward English I am not the first word is a real word.
  15. Why was the retinoblastoma (Rb) left out of this review as most tumors have mutations in either p53 or Rb or both.
  16. The data on Snail2 in line 146-147 is from a group of 20 dogs and the numbers are too small to be significant and you should indicate this. You do need to apply critical review skills in a review and not taking everything at face value.
  17. Line 150- it should “metastases” not “metastasis”
  18. Line 154- one of the “in vitro” non italicized and it is “scratch” not “scrath”
  19. Line 231- it is “northern and western blots”. They are not proper nouns. Please correct this throughout the paper.
  20. Line 232-it should be IHC as you have introduced the term previously? This also applies in TGFb in line 253.
  21. Line 234- “mild” is meaningless in this context. If you are talking about IHC staining it could be “weak” or “moderate”. Also replace with “immunolabelling” with “immunostaining”
  22. Line 235- “enhaced”?
  23. Line 260- what about a reference after the list of cell lines.
  24. Line 310 -what about a reference after the list of cell lines.
  25. Line 342- was this collagen type 1? Also replace “determined” with “demonstrated”
  26. Line 365-it is protein kinase C (PKC) -again it is not a proper noun.
  27. Line 375- how about a reference after “cell line”.
  28. Line 462 “dogs” to replace “canines”
  29. Line 517- what does “converged”mean?

Author Response

Dear Rewiever,

thank you very much or your  detailed review and  comments. Please find below the answers to each comment. We rewrote the article taking into consideration all your suggestions. We put the detail information on cell lines and methodology into new table 1, in which we also added the information on the steps of metastatic cascade. The article is shortened according to the suggestion of the other Reviewer. The article was one more time send  for English and scientific editing.

Response to your kind comments:

Comments and Suggestions for Authors

The authors have reviewed some of the molecular mechanisms of OS metastasis. There are numerous problems with the English and its usage.

 Comment 1:  Table 1 is almost unreadable. May be lines between the each of Proteins or genes would help. If there is a gene like p63 it must be in italics and that would mean that you do not have to stipulate gene.

Answer 1: We changed Table 1 into a new one, adding lines between the each of protein and gene. We change p63 into italics and removed word "gene"-> "p63" within the manuscript (lines: 96, 97).

Comment 2: Figure 1- I am not sure why this Figure is here. Upper panel- If you are going to mention the non-stochastic pathogenesis of metastases then tie it in with your list of factors/ genes. Which factor/gene relate to each of these steps. Also quote on of the Fidler papers -may be his 2003 Nature Reviews Cancer review examining ‘seed and soil’ hypothesis and then tie into Table 1. Lower panel- what is it saying? If it does not make a point then remove it. Location of organs is meaningless.

Answer 2: We changed the Figure 1 into a new one, adding the names of genes/proteins included in a particular step of metastatic cascade, and we removed the organ distribution of metastasis, as indicated .

We changed Table 1 into a new one, adding a column with steps of metastasis cascade, as indicated by both Reviewers.

Comment 3: Need to check the spacing as words run together like in line 106.

Answer 3: The article was sent once more for English language correction to be checked for grammatical and spelling errors. We have already realized that frequent missing spaces appeared on some, but not every, computer probably due to the fact that one of the Authors used an older version of MS Word while preparing the manuscript. We kindly apologize for the inconvenience with the words merged, but this was not seen on the final version of the article on the computer of the corresponding Author, and we strongly hope that this technical problem will not appear in the reviewed version of the manuscript.

Comment 4: First line of the abstract- “canines” has to be replaced by “dogs” This is also found in other places in this manuscript. Please correct this.

Answer 4: We changed word "canines" to "dogs" within the whole manuscript as indicated (lines 7, 148, 455).

Comment 5: “Mice” must replace “murines” in lines 57 and 58

Answer 5: We replaced word "mice" to "murines" (lines55, 56, 58, 110, 115, 224, 336).

Comment 6: Many of the sentences in the manuscript are over 30 words long and that results in the meaning being lost. This has happened throughout the manuscript and this is a basic mistake.

Answer 6: We shortened the sentence in the manuscript and also sent the article for English correction to shorten the sentences.

Comment 7: Also there is inconsistent use of italics in “in vivo” and ”in vitro”. Just choose one and stick to it.

Answer 7: We used italics as suggested by The Reviewer and changed it within the whole manuscript. Lines 209, 298, 409, 584 words "in vitro" were italicized. Lines 357, 409, 564, 476 words "in vivo" were also italicized.

Comment 8: Line 28 remove “a high”. Aggressiveness is enough and then it is “a propensity”

Answer 8: Line 29 – We removed words "a high" and added  "a" before the word "propensity," as indicated.

Comment 9: Line 31 it is “diagnosis”

Answer 9: Line 32- We changed "diagnosing" into "diagnosis," as indicated.

Comment 10: Line 33 insert “probably” before “due to” as there all the dogs were not autopsied.

Answer 10: Line 34 - We added word "probably" before "due to" as indicated.

Comment 11: Line 61 the language is too extravagant. Remove “definitively”.

Answer 11: Line 90 - We removed word "definitively," as indicated.

Comment 12: Lines 67- remove sentence starting “in the European..” Rewrite next sentence.

Answer 12: Line 56 - We removed the sentence started with "In the European Union..." and the next one according to suggestions of both Reviewers.

Comment 13: Line 94- put in an original reference for p63!

Answer 13: Line 133 - We added reference 21.

Comment 14: Line 104- rewrite the sentence- really awkward English I am not the first word is a real word.

Answer 14:We send the manuscript one more time for English and scientific editing.

Comment 15: Why was the retinoblastoma (Rb) left out of this review as most tumors have mutations in either p53 or Rb or both.

Answer 15: We are not sure where can we add the information on p53 mutations in retinoblastoma as the review article is on osteosarcoma.

Comment 16: The data on Snail2 in line 146-147 is from a group of 20 dogs and the numbers are too small to be significant and you should indicate this. You do need to apply critical review skills in a review and not taking everything at face value.

Answer 16: We totally agree with the Reviewer. Lines 147-149 "It should be considered that the study was performed on small number of dogs and archived material was used for IHC, which may limit the results."

Comment 17: Line 150- it should “metastases” not “metastasis”

Answer 17: We changed "metastases" into "metastasis"

Comment 18: Line 154- one of the “in vitro” non italicized and it is “scratch” not “scrath”

Answer 18: We italicized the words "in vitro" and changed "scrath" into "scratch"

Comment 19: Line 231- it is “northern and western blots”. They are not proper nouns. Please correct this throughout the paper.

Answer 19: This correction has been made. Thank you for the kind correction.

Comment 20: Line 232-it should be IHC as you have introduced the term previously? This also applies in TGFb in line 253.

Answer 20: We corrected it for both IHC and TGFbeta

Comment 21: Line 234- “mild” is meaningless in this context. If you are talking about IHC staining it could be “weak” or “moderate”. Also replace with “immunolabelling” with “immunostaining”

Answer 21: We agree with the Reviewer; however, this part of the manuscript was deleted according to the Reviewer 2 suggestions that it is not related to canine osteosarcoma metastasis. We changed word "immunolabelling" into "immunostaining" (line 228).

Comment 22: Line 235- “enhaced”?

Answer 22: We corrected the spelling for "enhanced" (line 154)

Comment 23: Line 260- what about a reference after the list of cell lines.

Answer 23: We removed the list of cell lines in the main part of the manuscript according to Reviewer 2 suggestions, but we add it into new Table one with reference.

Comment 24: Line 310 -what about a reference after the list of cell lines.

Answer 24: We added the reference [34] (line 324) at the end of the sentence, however, we deleted the list of cell lines in the main part of the manuscript according to Reviewer 2 suggestions, but we add it into new Table one with reference.

Comment 25: Line 342- was this collagen type 1? Also replace “determined” with “demonstrated”

Answer 25: We replaced the word "determined" with "demonstrated" as indicated (line 323)

Comment 26: Line 365-it is protein kinase C (PKC) -again it is not a proper noun.

Answer 26: This correction has been made (line 348)

Comment 27: Line 375- how about a reference after “cell line”.

Answer 27: Lines351 we added reference [81] .

Comment 28: Line 462 “dogs” to replace “canines”

Answer 28: We replaced the word "canines" into "dogs."

Comment 29: Line 517- what does “converged” mean?

Answer 29: We removed this sentence according to the suggestions of the Reviewer 1.

I hope the revised article will meet your criteria to be published in the International Journal of Molecular Sciences.

KInd regards,

Katarzyna Zabielska-Koczywąs

Reviewer 2 Report

In the presented review of “Molecular mechanisms of canine osteosarcoma metastasis”, the authors compile data from somewhat recent articles on the topic of canine osteosarcoma (OSA) metastisis.  The manuscript focuses on presumably metastasis related genes p63, STAT3, Snail2, Vimentin, VEGF, IGF-1, TGFb, HGF-SF/c-MET, EGFR, Integrins, CD147, Collagen, Ezrin/PKC, p-ERM, YAP, TAZ,  MiR-9, MiR-34a,  Annexins, TF, MMP-2, and MMP-9.  The manuscript is well researched and the authors identified most of the limited number of publications available that cover the topic of canine OSA metastasis.  The review manuscript, however, requires major editing. 

  1. Format/Spelling/Grammar

There are extensive formatting (missing spaces), spelling, and grammatical errors that make the article difficult to read.  There are 2 or more spelling errors in every paragraph and frequent missing spaces that result in words merged together.

The manuscript would benefit from more abbreviations.  Perhaps use “OSA” instead of “osteosarcoma”.  Also “MA” for migration assays.  There are also situations where abbreviations are introduced several times – the MMPs are an example (line 135, 171, 341, 479, 484).  Once “MMP” is introduced, it doesn’t need to be introduced again for each MMP#.  Reintroducing it in the section title of line 472, however, is appropriate. 

  1. Introduction

The introduction is mostly paraphrasing from previously published reviews.  While this is to be expected, the authors also cite review papers when presenting statistics regarding OSA in dogs, rather than the original source.  For example, the authors use reference #3 for the estimated incidence of OSA in dogs of 13.8/10,000.  Ref #3 gets that statistic from 2 other references from 2001 (small animal clinical oncology text book) and 2007 (US pet owner demographics sourcebook).  Ref #3 admits in their text that there is no consistent method for reporting cancer in dogs and that the reported 13.9/10,000 OSA incidence in dogs is just an estimate based on data from the 1990s (30 years ago).  The authors of this review manuscript should similarly acknowledge that there is limited data on these statistics and also cite the original source rather than the review paper.  Or perhaps find more recent papers that have updated statistics?

The authors do not introduce any of the genes/proteins/pathways in the introduction.  A few are mentioned in the abstract, but none are introduced in the introduction.  It would be important to introduce some of the genes/proteins in the context of the metastatic cascade.  Table 1 presents a list of these genes/proteins and the studies they are in, which is appropriate.  The authors should either add an additional column or create a new table (recommended) that indicates how each of these genes are involved in the metastatic cascade.  The authors also do not explain WHY they chose to focus their review on these specific genes and not the dozens of other genes cancer/metastasis related genes/proteins/pathways.

Figure 1 is too basic and not very informative.  The pictures lack identifying labels, such as vasculature, stroma, etc.  Including more detailed images of vasculature membranes as well as adding in the specific genes/proteins that play a role in each process would be much more informative.

  1. Citations

Reference #2 is not appropriate for lines 22-23 in the introduction.  There is no data regarding  OSA accounting for a specific percentage of malignant bone tumors in that article.

The citations are missing or placed late in the text in many circumstances.  Frequently a new study is introduced at the beginning or middle of a paragraph using the notation “Smith et al.”, but no citation is given until the end of the paragraph.  The citation should be present in the sentence where the study is introduced.  For example, line 128-129, “Fossey et al. established the activation of STAT3…” but no citation is found until 5 sentences later on line 139 – ref #19.   Instead, it should read “Fossey et al. [19] established the activation…”

In addition, it is not always clear which study the authors are referencing, especially when they jump between assays that would seem to come from separate studies, but actually come from the same study.  The authors should use terms like “In the same study…” or “the same investigators also found that…” when presenting different data from the same study to indicate that they are still referencing the same study and not a different study.

For example, Line 213 – “In veterinary medicine, treating cells with IGF-1 increased invasiveness of the Grey cell line, but not Abrams and D17 cell lines, using the Matrigel invasion assay [36], which suggest that the role of IGF-1 in osteosarcoma invasive phenotype is cell line-dependent. Furthermore, an association of IGF-1 expression in canine osteosarcoma cell lines (D17, Grey, and the Abrams cell lines) with tumorigenesis and the occurence of metastases was evaluated in immunodeficient murine models.”    It’s not clear that the in vitro invasion assay and the in vivo mouse model experiments were from the same study.

  1. Too many method details.

The authors unnecessarily describe method details from the studies they are referencing.  In most cases, there is no need to list every cell line that was used in the study.  This is done throughout the manuscript and should be edited out.  That information could go into a new column on Table 1.  Instead, they can just say “canine OSA cell lines” in most cases (some exceptions mentioned later). 

For example:  Line 213: “In veterinary medicine, treating cells with IGF-1 increased invasiveness of the Grey cell line, but not Abrams and D17 cell lines, using the Matrigel invasion assay [36], which suggest that the role of IGF-1 in osteosarcoma invasive phenotype is cell line-dependent.”    Can be shortened to:  “In one study, treating canine OSA cells with IGF-1 increased invasiveness of some, but not all cell lines [36], which suggest that the role of IGF-1 in osteosarcoma invasive phenotype is cell line-dependent.”

In this way, the main point of the sentence, “that the role of IGF-1 in osteosarcoma invasive phenotype is cell line-dependent” is clear and including the names of the cell lines and the specific migration assays used does not benefit the sentence.  Similarly, the authors do not need to describe the exact details of every migration and other assays used in their references.

Another example:  Line 484 – “Lana et al.detected expression of MMP-2 (gelatinase A) and matrix metalloproteinase 9 (MMP-9) (gelatinase B) in all 30 samples derived from spontaneously occured high-grade osteosarcoma tumors in dogs and most of the unaffected nearby stromal tissues (used as a control), analyzed using gelatin zymograhpy. In comparison to stromal tissues, the expression of MMP-2 and MMP-9 was higher (p<0.001) in the tumor tissues[97].”    This can be shortened to “Lana et al. [97] detected significantly higher MMP2 and MMP9 expression in 30 high grade canine osteosarcoma samples relative to nearby stromal control tissue.”

In some situations, however, additional information regarding the cell lines is needed to provide context.   This is especially true when cell lines are described as “metastatic” and “non-metastatic” such as the POS and HMPOS cell lines used in the frequently cited reference #29.  It is important to note that the POS cell line was derived from a primary spontaneous canine femoral OSA and the HMPOS was derived from POS after serially implantation into nude mouse lungs (Original paper - Barroga 1999, J Vet Med Sci).  This is an important distinction because HMPOS was artificially generated as a metastatic cell line, rather than a cell line derived from a spontaneous metastatic OSA.  This is also important to note because the authors use that same limited study to support a role for Vimentin, CD147, and Annexin in canine OSA metastasis.  The authors should cite the original POS/HMPOS paper and also acknowledge the limitations of this data as being an artificial model, not spontaneous, and also only 1 cell line in 1 study.

  1. Too much summary and not enough review.

Much of this manuscript is spent summarizing the methods and results of various canine OSA studies, without providing much discussion of how the data from all these studies creates a larger picture of canine metastatic OSA.   In addition, the title suggests the manuscript will focus on the molecular mechanisms involved in canine OSA metastasis, but much of the presented data is not specific to metastasis, but rather canine OSA in general.

The authors focus much of the introduction on the “metastatic cascade” and the various steps involved.  They even provide a basic figure to highlight those steps.  I would have expected the rest of the manuscript to then discuss all of the molecular mechanisms (genes/proteins/pathways) in the context of the various steps of the metastatic cascade.  The manuscript, however, proceeds to introduce each of the gene/proteins individually, many times with minimal or no context to the metastatic cascade.  Each paragraph could be a stand-alone abstract, and the authors fail to bring any of it together in the larger context of metastasis. Furthermore, only a few of the described studies actually show some type of mechanistic or functional significance of the investigated gene/protein with respect to metastasis. 

For example: Line 239:  Maniscalco et al. demonstated a shorter survival of dogs with appendicular osteosarcoma with high IGFR-1 immunoexpression as opposed to those with low IGFR-1 immunoexpression, using univariate and multivariate analyses and displayed no differences in DFI[17].  – There is nothing about metastasis here.  This is one of MANY examples.

  1. Additional Line-Specific comments:

Line 14 – format “Snail,ezrin” – “Snail2, ezrin”

Line 13 – don't need the “e.g”, the “including” covers that and the e.g. is redundant.

Line 14 – to many “ands” – need commas and only 1x “and”

Line 54 – “The perfect in vivo model to study preclinical metastasis should produce metastases within a few months, be immunocompetent with the tumor specimen, and be orthotropic[9]” .  Do the authors mean a perfect model for human OSA or for canine OSA, or both?  Also – spelling – “orthotopic” not “orthotropic”.  This sentence is paraphrased from another review paper and lacks context here.

Line 58 – “Moreover, for canine osteosarcoma, the technique of tumor implantation excludes the probability of osteosarcoma cells seeding into pulmonary blood vessels and exemplifies all of the metastatic cascade steps[14], which is definitively advantageous over the commonly used injection of osteo-sarcoma cell suspensions into the femur or tibia of murines.”  There is no context for this sentence and it is therefore confusing.  What is the “tumor implantation technique” that the authors are referring to?  How does “excluding the probability of OSA cells seeding into pulmonary blood vessels” exemplify the metastatic cascade?  This seems to have been taken out of context from another review article and paraphrased.

Line 62 – “No association between intra-venous (IV) and intratibial (IT) transplantation sites and the development of lung metastases were observed, but there is clear evidence of the cell line-dependent metastatic potential of canine osteosarcoma.”  Again, no context for this sentence.  What are the authors observing?  This seems like it was pulled from another reference without context, and without a clear citation.  This makes no sense.

Line 65 – The authors abruptly change topics mid-paragraph to animal welfare and the “3Rs”.  This seems completely out of place and incongruent.

Section 2.1 p63 gene.   Formatting – spaces are missing in several lines (line 96 “toa”, line 101 “performed.Investigators”, line 103, line 103, line 106… many more throughout the manuscript.

Line 97 – Cam et al. is missing the actual citation, which does not appear until line 110.

Line 141 – italicize “in vitro”

Line 144-147 - Regarding Snail2 expression in high, intermediate, and low grade osteosarcomas – there is no citation here.  Reference?

Line 149-152:  “Appendicular long bone osteosarcoma in dogs metastasizes in 80-90% of cases[1], while skull osteosarcoma metastasis occur less frequently, and according to the authors, this fact may suggest that Snail2 could play a role in process of cell migration[25].”

 -  It is not clear which citation is being used for which text here.  The authors state “according to the authors”, but it’s not clear which authors or which citation (#1 or #25). Do the authors of ref #25 cite research from Ref #1?  The sentence is confusing.

Line 153 – “may implicate the results”.  Do the authors mean “may limit the results”, or something similar?

Line 154 – “Scrath assay” – “Scratch assay”?

Line 156 – “demonstated”

Line 156 “comparision”

Line 155-156: “(to gain over-expression of Snail2)”.  No need for parenthesis.

Line 158 – “Snail2 does not” – Snail2 DID not”

Line 187 – I disagree with the authors comment regarding the human OSA cell line comparison in the referenced study (#29).  It did not show an “identical association” as the canine cell lines.  The difference in vimentin in the canine OSA cell lines was over 3-fold (216% increase), while the difference between the human cell lines was only 1.2-fold (20% increase).  And there was no statistical analysis performed in that study.

Line 203 – “Depletion of ΔNp63 in canine D17 osteosarcoma cells resulted in a decreased level of VEGF and a noticeable reduction of endothelial tube formation in transwell endothelial tube formation assays (using human umbilical vein endothelial cells).”  This sentence is confusing. The tube formation was not performed in canine cells?  How is this significant to canine OSA metastasis?

Line 208 - IGF1 section.  The 2 paragraphs are very similar and should be consolidated into 1 paragraph.  The paragraphs are difficult to understand with long sentences with lots of parenthesis.  There is too much detail about the specific methods used in each of these studies for a review paper.  These paragraphs attempt to summarize the methods and results of 2 studies, but no discussion of how the results of both studies complement each other or don’t.   

Paragraph 227.  The paragraph is incoherent and jumps between topics without warning.  It starts out introducing the subunits of IGFR-1, then the next sentence says IGFR-1 is associated with poor outcome in human OSA, then the next sentence is about results from canine OSA studies.  In addition, the sentence about the canine OSA cell lines is very confusing:

“Investigators established an expression of IGFR-1 in canine osteosarcoma cell lines analysed  with Northern and Western Blot assays (Abrams, Grey, D-17, SK-KOSA, MC-KOSA, and BW-KOSA cell lines)[36,44] and canine osteosarcoma tissues examined by immunohistochemistry (24 of 34 samples derived from spontaneously occuring osteosarcoma in dogs), which presented mild (n=8) and strong (n=16) cytoplasmic immunolabelling[17].”   - There are 3 different assays described here.  Did the Northern and Western Blot assays show something significant?  What do you mean “established an expression?”  This sentence is extremely confusing.

Line 236 – “Orthotopic (distal femur) injection of immunodeficient mice with solid tumors displayed spontaneous metastasis following amputation, making the presented model an important spontaneous canine osteosarcoma model to further investigate the role of the IGF-1 system in osteosarcoma metastasis.”  - it is not clear which study this data is from.  Where did the solid tumors come from that were injected?  What is the presented model? What does this have to do with the molecular mechanisms of metastasis?

Line 239-242 – It is not clear how this data relates to metastasis.

Line 245 - 2.5.3 TGFb section.  The authors present studies that suggest TGFb is “not characteristic of the malignant phenotype” of OSA and the blocking a “Smad-dependent TGFB-mediated pathway” decreased migration in some canine OSA cell lines, but not others.  Furthermore, that complete blocking of “canonical and non-canonical signaling” did not result in inhibition of cell motility.  There is no discussion of what those results mean for the role of TGFb in canine OSA metastasis.  The last sentence of that paragraph “Presumably, substitutive signaling pathways, like c-Met, have the ability to sustain OSA cell migration[48].” Seems random and lacks context.  What is the relationship between c-Met and TGFb?  Are the authors suggesting that TGFb is not important in canine OSA metastasis?

Line 420 – “Fenger et.al evaluated up-regulation of miR-9 in canine osteosarcoma cell lines (OSA8 and OSA16) and in primary osteosarcoma tumors, in comparison to normal canine osteoblasts[86].”  What did Fenger et al discover regarding upregulation?  Do the authors mean “Fenger et al. observed up-regulation of miR-19?”

Line 483 – MMP-2 and MMP-9 section.  The entire paragraph regarding MMP-2 and MMP-9 basically states that some studies have found increased levels of MMP2 and 9 in canine primary osteosarcomas, but there is no data about those proteins in OSA metastasis.   Becuase there is not much data to present regarding MMPs and metastasis, there is no reason to devote a whole paragraph to this.  The whole paragraph could be added to the previous paragraph and shortened to 1-2 sentences. 

Line 499 – “Unfortunately, authors found no studies of MMP expression in…” Which authors?  Are the authors of this review talking about themselves in the 3rd person?

Table 1.  This is an appropriate location to list the methods and cell lines/ samples used in all of the cited references.  There is no need to include all of this information in the text of the manuscript.

Line 506 - Conclusions section – There is nothing here that ties all the introduced genes/proteins/pathways together in the context of metastasis, or any context.

Line 153 -  “The strongest evidence evaluating both in vitro and in vivo studies presented for: ΔNp63, Snail2, IGF-1, IGFR-1, EGFR, 514 HGF-SF, CD147, ezrin, p-ERM, miR-9 and miR-34a.”   - Is this just a statement?  Is there supposed to be more to this sentence?

Line 515 – “The results of the in vitro and in vivo tests converged on these factors, suggesting their role in osteosarcoma cell invasion, migration, and formation at secondary sites as important steps in the metastatic cascade.”  How do the results of these studies “converge”?  What are the factors that they are converging on?  I don't understand what this means.

Author Response

Dear Rewiever,

thank you very much for your detailed review and great comments. Please find below the answers to each comment. We rewrote the article taking into consideration all your suggestions. The article is shortened, we put the detail information on cell lines and methodology into new table 1, in which we also added the information on the steps of metastatic cascade. The article was one more time send  for English and scientific editing.

Response to your kind comments: 

  • Format/Spelling/Grammar

Comment 1: There are extensive formatting (missing spaces), spelling, and grammatical errors that make the article difficult to read.  There are 2 or more spelling errors in every paragraph and frequent missing spaces that result in words merged together.

Answer 1: The article was sent once more time for English language correction to be checked for grammatical and spelling errors. We also realized that frequent missing spaces appeared on some, but not every version of the submitted manuscript. It seems these errors result from the fact that one of the Author’s used an older version of MS Word while preparing the manuscript. We kindly apologize for the inconvenience with the merged words, but this was not seen on the final version of the article on the computer of the corresponding Author. We strongly hope that this technical problem will not appear in this updated, revised version of the manuscript.

Comment 2: The manuscript would benefit from more abbreviations.  Perhaps use “OSA” instead of “osteosarcoma”.  Also “MA” for migration assays.  There are also situations where abbreviations are introduced several times – the MMPs are an example (line 135, 171, 341, 479, 484).  Once “MMP” is introduced, it doesn’t need to be introduced again for each MMP#.  Reintroducing it in the section title of line 472, however, is appropriate. 

Answer 2: We added more abbreviations as indicated by the Reviewer. We changed osteosarcoma to OSA and wound healing assay to WHA within the whole manuscript. We also corrected the use of abbreviations for MMPs and once more checked the correct use of abbreviations within the entire manuscript.

  • Introduction

Comment 3: The introduction is mostly paraphrasing from previously published reviews.  While this is to be expected, the authors also cite review papers when presenting statistics regarding OSA in dogs, rather than the original source.  For example, the authors use reference #3 for the estimated incidence of OSA in dogs of 13.8/10,000.  Ref #3 gets that statistic from 2 other references from 2001 (small animal clinical oncology text book) and 2007 (US pet owner demographics sourcebook).  Ref #3 admits in their text that there is no consistent method for reporting cancer in dogs and that the reported 13.9/10,000 OSA incidence in dogs is just an estimate based on data from the 1990s (30 years ago).  The authors of this review manuscript should similarly acknowledge that there is limited data on these statistics and also cite the original source rather than the review paper.  Or perhaps find more recent papers that have updated statistics?

Answer 3: We changed the estimated incidence of OSA to 5.6-13.8/10,000, as it is stated in the original articles on incidence rates of canine neoplasia including OSA, which is also 5.6/10,000 (reference 3 and 4). We also leave article 5 as a citation for the 13.8/10,000 estimated incidence rate, as the original file for it (US pet owner demographics sourcebook) is now unavailable. We also added information on limited data on the statistics of canine OSA and the fact that there are no consistent methods of reporting cancers in dogs (lines 22-24).

Comment 4: The authors do not introduce any of the genes/proteins/pathways in the introduction.  A few are mentioned in the abstract, but none are introduced in the introduction.  It would be important to introduce some of the genes/proteins in the context of the metastatic cascade.  Table 1 presents a list of these genes/proteins and the studies they are in, which is appropriate.  The authors should either add an additional column or create a new table (recommended) that indicates how each of these genes are involved in the metastatic cascade.  The authors also do not explain WHY they chose to focus their review on these specific genes and not the dozens of other genes cancer/metastasis related genes/proteins/pathways.

Answer 4: We added information on which genes/proteins will be discussed further in the article as suggested by the Reviewer (lines 75-78). We created a new Table 1, as suggested by the Reviewer, in which we added the column titled “Step(s) of the metastatic cascade,” containing information for each gene/protein involved. We also added the section "Search methodology" (lines 82-92) where we demonstrated how we performed the literature search. This section explains why those specific genes/proteins are included in the present review.

Comment 5: Figure 1 is too basic and not very informative.  The pictures lack identifying labels, such as vasculature, stroma, etc.  Including more detailed images of vasculature membranes as well as adding in the specific genes/proteins that play a role in each process would be much more informative.

Answer 5: We decided to remove Figure 1 as we agree with both Reviewers that it is not informative. 

Citations

Comment 6: Reference #2 is not appropriate for lines 22-23 in the introduction.  There is no data regarding  OSA accounting for a specific percentage of malignant bone tumors in that article.

Answer 6: We changed the references for the specific percentage of malignant bone tumors in dogs (new reference 3, 4 and 5) (line 22).

Comment 7: The citations are missing or placed late in the text in many circumstances.  Frequently a new study is introduced at the beginning or middle of a paragraph using the notation “Smith et al.”, but no citation is given until the end of the paragraph.  The citation should be present in the sentence where the study is introduced.  For example, line 128-129, “Fossey et al. established the activation of STAT3…” but no citation is found until 5 sentences later on line 139 – ref #19.   Instead, it should read “Fossey et al. [19] established the activation…”

Answer 7: We added citations next to the Authors names as indicated:

line 100 Cam et al. [22]

line 127 Fossey et al. [20]

line 146 Sharil et al. [26]

line 171 Lana et al. [31]

line 173 Loukopoulos et al. [31]

line 175 Loukopoulos et al. [31

line 192 Roy et al. [34]

line 209 Cam et al.[22]

line 308 Roy et al. [34]

line 342 Jackson et al. [66]

line 417 Lopez et al. [98]

line 424 Lopez et al. [98]

line 440 Roy et al. [34]

line 45Stokol et al. [105] in the same study

Comment 8: In addition, it is not always clear which study the authors are referencing, especially when they jump between assays that would seem to come from separate studies, but actually come from the same study.  The authors should use terms like “In the same study…” or “the same investigators also found that…” when presenting different data from the same study to indicate that they are still referencing the same study and not a different study.

For example, Line 213 – “In veterinary medicine, treating cells with IGF-1 increased invasiveness of the Grey cell line, but not Abrams and D17 cell lines, using the Matrigel invasion assay [36], which suggest that the role of IGF-1 in osteosarcoma invasive phenotype is cell line-dependent. Furthermore, an association of IGF-1 expression in canine osteosarcoma cell lines (D17, Grey, and the Abrams cell lines) with tumorigenesis and the occurrence of metastases was evaluated in immunodeficient murine models.”    It’s not clear that the in vitro invasion assay and the in vivo mouse model experiments were from the same study.

Answer 8: We added to the reviewed version of the manuscript the information "in the same study" or "the same investigators also found that" where appropriate:

Lines 101-102 "in the same study" was aded.

Line 113  "in the same study" was added.

Line 124 "The investigators preformed further studies" was added.

Line 128  "in the same study" was added.

Line 160 "Further studies" was added.

Line 175 "In further investigations" was aded.

Line 315 "further studies" was added.

Line 403 "in the same study" was added.

Line 428 "in further studies" was added.

  • Too many method details.

Comment 9: The authors unnecessarily describe method details from the studies they are referencing.  In most cases, there is no need to list every cell line that was used in the study.  This is done throughout the manuscript and should be edited out.  That information could go into a new column on Table 1.  Instead, they can just say “canine OSA cell lines” in most cases (some exceptions mentioned later). 

For example:  Line 213: “In veterinary medicine, treating cells with IGF-1 increased invasiveness of the Grey cell line, but not Abrams and D17 cell lines, using the Matrigel invasion assay [36], which suggest that the role of IGF-1 in osteosarcoma invasive phenotype is cell line-dependent.”    Can be shortened to:  “In one study, treating canine OSA cells with IGF-1 increased invasiveness of some, but not all cell lines [36], which suggest that the role of IGF-1 in osteosarcoma invasive phenotype is cell line-dependent.”

In this way, the main point of the sentence, “that the role of IGF-1 in osteosarcoma invasive phenotype is cell line-dependent” is clear and including the names of the cell lines and the specific migration assays used does not benefit the sentence.  Similarly, the authors do not need to describe the exact details of every migration and other assays used in their references.

Another example:  Line 484 – “Lana et al.detected expression of MMP-2 (gelatinase A) and matrix metalloproteinase 9 (MMP-9) (gelatinase B) in all 30 samples derived from spontaneously occured high-grade osteosarcoma tumors in dogs and most of the unaffected nearby stromal tissues (used as a control), analyzed using gelatin zymograhpy. In comparison to stromal tissues, the expression of MMP-2 and MMP-9 was higher (p<0.001) in the tumor tissues[97].”    This can be shortened to “Lana et al. [97] detected significantly higher MMP2 and MMP9 expression in 30 high grade canine osteosarcoma samples relative to nearby stromal control tissue.”

In some situations, however, additional information regarding the cell lines is needed to provide context.   This is especially true when cell lines are described as “metastatic” and “non-metastatic” such as the POS and HMPOS cell lines used in the frequently cited reference #29.  It is important to note that the POS cell line was derived from a primary spontaneous canine femoral OSA and the HMPOS was derived from POS after serially implantation into nude mouse lungs (Original paper - Barroga 1999, J Vet Med Sci).  This is an important distinction because HMPOS was artificially generated as a metastatic cell line, rather than a cell line derived from a spontaneous metastatic OSA.  This is also important to note because the authors use that same limited study to support a role for Vimentin, CD147, and Annexin in canine OSA metastasis.  The authors should cite the original POS/HMPOS paper and also acknowledge the limitations of this data as being an artificial model, not spontaneous, and also only 1 cell line in 1 study.

Answer 9: We removed the details such as names of cell lines within the text, were appropriate, as indicated by the Reviewer:
Lines 100-101 We removed names of cell lines: "the Abram's cell line, OSA8, OSA16"

Line 101 We removed the detail description of methodology about p63 as indicated to avoid too many methodology details within the manuscript.

Line 110 We removed "OSCA16 and D17 cells"

Line 129 We removed names of cell lines: "OSA8, OSA11M, OSA16, OSA29, OSA32 and D17 cell lines".

Lin 136 We removed "OSCA8, OSCA16 and OSCA32"

Line 171 We shortened the sentence to, “Lana et al. [97] detected significantly higher MMP2 and MMP9 expression in 30 high grade canine osteosarcoma samples relative to nearby stromal control tissue.” as indicated by the Reviewer.

Line 212 We removed name of cell line:"D17".

Line 213 We removed "using human and umbilical vein endothelial cells".

Line 220 We removed names of cell lines: "the grey cell line, but not Abrams and D17 cell lines, using Matrigel invasion assay".

Line 226 We removed "D17, Grey, and the Abrams cell lines "Lung metastasis were visible within 6 weeks in all nude mice xenografted orthotopically (distal femur) with Abrams cells (euthanized and necropsied in 6 weeks after amputation of the limb, performed when the tumor reached 10mm in diameter" as it does not provide any information related to metastasis.

Line 279 We removed "COS31, HMPOS, POS, D17, KOS-001, KOS-002, KOS-003, KOS-004"

Line 296 we removed "SK-KOSA and BW-KOSA"

Lines 291-294 We added additional information about POS and HMPOS cell lines as indicated by the Reviewer, "(...)in two canine osteosarcoma cell lines (non-metastatic POS and artificially generated metastatic HMPOS [66]. However, a strong limitation of the study is that the study was performed only on two OSA cell lines: POS-derived from a primary spontaneous canine femoral OSA and the HMPOS, which was derived from POS after serially implantation into the lungs of nude mice.”

Lines 314 We changed the word "should" into "may potentially."

Lines 315-316 We added, "cell lines (especially primary metastatic cell lines) and many more.”

Line 317 We changed, "should be examined" into "are obligatory to confirm this hypothesis."

Line 332-333 We added, "Nevertheless, the strong limitation of the study is that the study was performed on only one metastatic OSA cell line (HMPOS)."

  • Too much summary and not enough review.

Comment 10: Much of this manuscript is spent summarizing the methods and results of various canine OSA studies, without providing much discussion of how the data from all these studies creates a larger picture of canine metastatic OSA.   In addition, the title suggests the manuscript will focus on the molecular mechanisms involved in canine OSA metastasis, but much of the presented data is not specific to metastasis, but rather canine OSA in general.

The authors focus much of the introduction on the “metastatic cascade” and the various steps involved.  They even provide a basic figure to highlight those steps.  I would have expected the rest of the manuscript to then discuss all of the molecular mechanisms (genes/proteins/pathways) in the context of the various steps of the metastatic cascade.  The manuscript, however, proceeds to introduce each of the gene/proteins individually, many times with minimal or no context to the metastatic cascade.  Each paragraph could be a stand-alone abstract, and the authors fail to bring any of it together in the larger context of metastasis. Furthermore, only a few of the described studies actually show some type of mechanistic or functional significance of the investigated gene/protein with respect to metastasis. 

For example: Line 239:  Maniscalco et al. demonstated a shorter survival of dogs with appendicular osteosarcoma with high IGFR-1 immunoexpression as opposed to those with low IGFR-1 immunoexpression, using univariate and multivariate analyses and displayed no differences in DFI[17].  – There is nothing about metastasis here.  This is one of MANY examples.

Answer 10: We removed the information which were only for primary OSA instead of metastatic OSA and we added additional information on the role of each gene/protein in metastatic cascade within the manuscript as well as in Table 1. Indeed, we totally agree with the Reviewer that the presented review article preferably should have much more detailed information on molecular mechanisms of canine osteosarcoma metastasis; however, the studies performed in veterinary medicine for osteosarcoma did not provide such detailed information on molecular mechanisms as it is for example in human osteosarcoma. As a result the point of the presented article is to show what has been done so far in context of canine osteosarcoma metastasis  and what should be further assessed to really answer the question about molecular mechanisms of canine OSA, as well as which direction the investigators should choose based on the results from the up-to-date performed research within this topic.

We removed the part of the text about IGFR1 immunoexpression and its' influence on DFI as indeed it has no relation to metastasis (line 228)

  • Additional Line-Specific comments:

Comment 11: Line 14 – format “Snail,ezrin” – “Snail2, ezrin”

Answer 11: Line 14 - We added a space between the words, and we sent the article once more time for formatting. As previously explained, unfortunately, the problem with missing free spaces appeared due to an old version of MS Word of one of the Authors of this article. We kindly apologize for it, as we did not realize the problem before submitting the manuscript.

Comment 12: Line 13 – don't need the “e.g”, the “including” covers that and the e.g. is redundant.

Answer 12: Line 13 - We removed e.g. as indicated.

Comment 13: Line 14 – to many “ands” – need commas and only 1x “and”

Answer 13: Line 14 - "and" is changed into "'," as indicated

Comment 14: Line 54 – “The perfect in vivo model to study preclinical metastasis should produce metastases within a few months, be immunocompetent with the tumor specimen, and be orthotropic[9]” .  Do the authors mean a perfect model for human OSA or for canine OSA, or both?  Also – spelling – “orthotopic” not “orthotropic”.  This sentence is paraphrased from another review paper and lacks context here.

Answer 14: In this particular sentence we  mean the in vivo metastatic model in general, no matter if it is human or canine osteosarcoma. However, in the next sentence we deleted the details the context for both human and canine osteosarcoma.(lines 54-59)

Line 54 - We corrected the spelling "orthotropic" into "orthotopic"

Comment 15: Line 58 – “Moreover, for canine osteosarcoma, the technique of tumor implantation excludes the probability of osteosarcoma cells seeding into pulmonary blood vessels and exemplifies all of the metastatic cascade steps[14], which is definitively advantageous over the commonly used injection of osteo-sarcoma cell suspensions into the femur or tibia of murines.”  There is no context for this sentence and it is therefore confusing.  What is the “tumor implantation technique” that the authors are referring to?  How does “excluding the probability of OSA cells seeding into pulmonary blood vessels” exemplify the metastatic cascade?  This seems to have been taken out of context from another review article and paraphrased.

Answer 15: We removed the part of the sentence about "technique of implementation" and rewrite the sentence to clarify it:

lines 56-60 "Moreover, the tumor implantation in both canine and human OSA is definitively more advantageous over the commonly used injections of OSA cell suspension into the femur or tibia of murines, as it exemplifies all of the metastatic cascade steps[16]."

Comment 16: Line 62 – “No association between intra-venous (IV) and intratibial (IT) transplantation sites and the development of lung metastases were observed, but there is clear evidence of the cell line-dependent metastatic potential of canine osteosarcoma.” Again, no context for this sentence.  What are the authors observing?  This seems like it was pulled from another reference without context, and without a clear citation.  This makes no sense.

Answer 16: We removed this information as indeed it was unnecessary in this review article (line 64).

Comment 17: Line 65 – The authors abruptly change topics mid-paragraph to animal welfare and the “3Rs”.  This seems completely out of place and incongruent.

Answer 17: We removed the information about the 3R guidelines as we completely agree with the Reviewer that it was unnecessary (line 64).

Comment 18: Section 2.1 p63 gene.   Formatting – spaces are missing in several lines (line 96 “toa”, line 101 “performed.Investigators”, line 103, line 103, line 106… many more throughout the manuscript.

Answer 18: Line 97 - we added "space" between the words "encodes" and "proteins"

Line 99 we added space between words "to" and "a"

Comment 19: 97 – Cam et al. is missing the actual citation, which does not appear until line 110.

Answer 19: Line 100 - We added reference after Cam et al. [22].

Comment 20: Line 141 – italicize “in vitro”

Answer 20: Lines 209, 298, 409, 584 words "in vitro" were italicized.

Lines 357, 409, 564, 476 words "in vivo" were also italicized.

Comment 21: Line 144-147 - Regarding Snail2 expression in high, intermediate, and low grade osteosarcomas – there is no citation here.  Reference?

Answer 21: Line 140 – We added the reference [25].

Comment 22: Line 149-152:  “Appendicular long bone osteosarcoma in dogs metastasizes in 80-90% of cases[1], while skull osteosarcoma metastasis occur less frequently, and according to the authors, this fact may suggest that Snail2 could play a role in process of cell migration[25].”

 -  It is not clear which citation is being used for which text here.  The authors state “according to the authors”, but it’s not clear which authors or which citation (#1 or #25). Do the authors of ref #25 cite research from Ref #1?  The sentence is confusing.

Answer 22: Line 143 - We added the citation [26] and the word "furthermore" "(...)and low grade (n=4) samples[25]. Furthermore,".

Line 144 We clarified the next sentence by adding "as it is well known that"

Line 146 We changed "the Authors" into "Sharili et al. [25]" to clarify who is the author if which statement.

Comment 23: Line 153 – “may implicate the results”.  Do the authors mean “may limit the results”, or something similar?

Answer 23: Line 149 We changed the word "implicate" into "limit" to clarify our intentions.

Comment 24: Line 154 – “Scrath assay” – “Scratch assay”?

Answer 24: Line 149 - we corrected the spelling mistake "scratch assay" into "scratch assay"

Comment 25: Line 156 – “demonstated”

Answer 25: Line 151 we corrected the spelling "demonstated" into "demonstrated"

Comment 26: Line 156 “comparision”

Answer 26: Line 151 we corrected the spelling "comparisoin" into "comparison"

Comment 27: Line 155-156: “(to gain over-expression of Snail2)”.  No need for parenthesis.

Answer 27: Line 152  We removed parenthesis.

Comment 28: Line 158 – “Snail2 does not” – Snail2 DID not”

Answer 28: Line 153  – We changed "does not" into "did not."

Comment 29: Line 187 – I disagree with the authors comment regarding the human OSA cell line comparison in the referenced study (#29).  It did not show an “identical association” as the canine cell lines.  The difference in vimentin in the canine OSA cell lines was over 3-fold (216% increase), while the difference between the human cell lines was only 1.2-fold (20% increase).  And there was no statistical analysis performed in that study.

Line 192 – We deleted the word "significant".v

Answer 29: Line 193 - We added "(over 3-fold)".

Line  194 - we changed "the identical" into "similar".

Line 196 – We added ", although the increase as only 1.2-fold higher".

Comment 30: Line 203 – “Depletion of ΔNp63 in canine D17 osteosarcoma cells resulted in a decreased level of VEGF and a noticeable reduction of endothelial tube formation in transwell endothelial tube formation assays (using human umbilical vein endothelial cells).”  This sentence is confusing. The tube formation was not performed in canine cells?  How is this significant to canine OSA metastasis?

Answer 30: Line 213 We deleted "(using human umbilical vein endothelial cells". We indicated that according to the Authors of the study, this finding is significant in context of metastasis as it may be promoting the process of angiogenesis [21].

Comment 31: Line 208 - IGF1 section.  The 2 paragraphs are very similar and should be consolidated into 1 paragraph.  The paragraphs are difficult to understand with long sentences with lots of parenthesis.  There is too much detail about the specific methods used in each of these studies for a review paper.  These paragraphs attempt to summarize the methods and results of 2 studies, but no discussion of how the results of both studies complement each other or don’t. 

Answer 31: Lines 215-232 (IGF1 section) – We consolidated paragraphs on IGF-1 and IGFR-1 into one section and shorten the sentences as well as deleted the unimportant ones with too detailed for the review article information on methodology used, as indicated by the Reviewer. We discussed that the studies do not complement each other, "Unfortunately, on canine samples only immunolabelling of IGFR-1 was performed, showing a shorter survival of dogs with appendicular OSA with high IGFR-1 immunoexpression, as opposed to those with low IGFR-1 immunoexpression, which does not provide any further information on the role of IGF-1 and IGFR-1 in metastatic cascade."

Comment 32: Paragraph 227.  The paragraph is incoherent and jumps between topics without warning.  It starts out introducing the subunits of IGFR-1, then the next sentence says IGFR-1 is associated with poor outcome in human OSA, then the next sentence is about results from canine OSA studies.  In addition, the sentence about the canine OSA cell lines is very confusing:

“Investigators established an expression of IGFR-1 in canine osteosarcoma cell lines analysed  with Northern and Western Blot assays (Abrams, Grey, D-17, SK-KOSA, MC-KOSA, and BW-KOSA cell lines)[36,44] and canine osteosarcoma tissues examined by immunohistochemistry (24 of 34 samples derived from spontaneously occuring osteosarcoma in dogs), which presented mild (n=8) and strong (n=16) cytoplasmic immunolabelling[17].”   - There are 3 different assays described here.  Did the Northern and Western Blot assays show something significant?  What do you mean “established an expression?”  This sentence is extremely confusing.

Answer 32: We deleted this information according to the earlier comment of the Reviewer that it is not connected with the metastatic cascade.

Comment 33: Line 236 – “Orthotopic (distal femur) injection of immunodeficient mice with solid tumors displayed spontaneous metastasis following amputation, making the presented model an important spontaneous canine osteosarcoma model to further investigate the role of the IGF-1 system in osteosarcoma metastasis.”  - it is not clear which study this data is from.  Where did the solid tumors come from that were injected?  What is the presented model? What does this have to do with the molecular mechanisms of metastasis?

Answer 33: We removed this information according to the earlier comment of the Reviewer that it is not connected with the metastatic cascade.

Comment 34: Line 239-242 – It is not clear how this data relates to metastasis.

Answer 34: We removed this information according to the earlier comment of the Reviewer that it is not connected with the metastatic cascade.

Comment 35: Line 245 - 2.5.3 TGFb section.  The authors present studies that suggest TGFb is “not characteristic of the malignant phenotype” of OSA and the blocking a “Smad-dependent TGFB-mediated pathway” decreased migration in some canine OSA cell lines, but not others.  Furthermore, that complete blocking of “canonical and non-canonical signaling” did not result in inhibition of cell motility.  There is no discussion of what those results mean for the role of TGFb in canine OSA metastasis.  The last sentence of that paragraph “Presumably, substitutive signaling pathways, like c-Met, have the ability to sustain OSA cell migration[48].” Seems random and lacks context.  What is the relationship between c-Met and TGFb?  Are the authors suggesting that TGFb is not important in canine OSA metastasis?

Answer 35: We made changes on lines 233-250.

Comment 36: Line 420 – “Fenger et.al evaluated up-regulation of miR-9 in canine osteosarcoma cell lines (OSA8 and OSA16) and in primary osteosarcoma tumors, in comparison to normal canine osteoblasts[86].”  What did Fenger et al discover regarding upregulation?  Do the authors mean “Fenger et al. observed up-regulation of miR-19?”

Answer 36: Line 417 --- We changed word "evaluated" to "observed" to clarify the meaning of the sentence.

Comment 37: Line 483 – MMP-2 and MMP-9 section.  The entire paragraph regarding MMP-2 and MMP-9 basically states that some studies have found increased levels of MMP2 and 9 in canine primary osteosarcomas, but there is no data about those proteins in OSA metastasis.  Because there is not much data to present regarding MMPs and metastasis, there is no reason to devote a whole paragraph to this.  The whole paragraph could be added to the previous paragraph and shortened to 1-2 sentences. 

Answer 37: We shortened the paragraph about MMPs and added it to the previous paragraph - paragraph 3.3 (lines 163-182).

Comment 38: Line 499 – “Unfortunately, authors found no studies of MMP expression in…” Which authors?  Are the authors of this review talking about themselves in the 3rd person?

Answer 38: Line 177 – We changed word "authors" into "we".

Comment 39: Table 1.  This is an appropriate location to list the methods and cell lines/ samples used in all of the cited references.  There is no need to include all of this information in the text of the manuscript.

Answer 39: We removed the names of cell lines in most cases within the manuscript, except for the places where the Reviewer or we believe it is necessary to point out the name of the cell lines as e.g., in case of POS/HMPOS. We add a new Table 1 as indicated by the Reviewer, and we add column (cell lines/samples as well as column step of metastatic cascade.

Comment 40: Line 506 - Conclusions section – There is nothing here that ties all the introduced genes/proteins/pathways together in the context of metastasis, or any context.

Answer 40: We rewrote the conclusion section indicating which genes/proteins have the strongest evidence, based on the reviewed studies, to take part in either migration, invasiveness or both steps of metastatic cascade (lines 469-486)

Comment 41: Line 153 -  “The strongest evidence evaluating both in vitro and in vivo studies presented for: ΔNp63, Snail2, IGF-1, IGFR-1, EGFR, 514 HGF-SF, CD147, ezrin, p-ERM, miR-9 and miR-34a.”   - Is this just a statement?  Is there supposed to be more to this sentence?

Answer 41: This is a sentence to point out the proteins that were evaluated both in vitro and in vivo for their potential role in canine OSA metastasis.

Comment 42: Line 515 – “The results of the in vitro and in vivo tests converged on these factors, suggesting their role in osteosarcoma cell invasion, migration, and formation at secondary sites as important steps in the metastatic cascade.”  How do the results of these studies “converge”?  What are the factors that they are converging on?  I don't understand what this means.

Answer 42: We removed this sentence.

I hope the revised article will meet your criteria to be published in the International Journal of Molecular Sciences.

Kind regards,

Katarzyna Zabielska-Koczywąs

Round 2

Reviewer 2 Report

The authors have addressed the majority of my comments adequately.  The manuscript has improved significantly, although several more moderate edits are needed prior to publication.  Please see my comments below. 

I thank the authors for this contribution to to the field of canine metastatic osteosarcoma research.

Line Comments:

Figure 1 – The author’s mention that this figure was removed, but the figure appears to still be present in the re-write. 

Line 65-70 – “Moreover, the tumor implantation in both canine and human OSA is advantageous over the commonly used injection of OSA cell suspensions into the femur or tibia of mice…”  The way it is written, the authors suggest that injecting humans and dogs with tumor cells is advantageous to injecting mice with OSA cells.  The word “implantation” suggests tumors are being injected into humans and dogs.    Do the authors mean “…the spontaneous tumor growth of both canine and human OSA is advantageous….”?  Or are the authors trying to describe a different method of implanting human and canine OSA cells into mice that differs from the method of injecting them into the femur or tibia.  It is not clear what the authors are trying to say here.

Line 78 – Remove the “:”

Line 87 – “Monitoring of in vivo macrometastases is performed with non-invasive imaging methods such as MRI or bioluminescence.”   - Do the authors have a reference for this statement?  X-rays are standard protocol for monitoring lung metastasis in dogs, not MRI or bioluminescence. 

Line 88 – “Both macro and micrometastases can be detected by other quantitative measurements of their burdens and histological analysis” – What other quantitative measurements are the authors referring to here?  Reference?  What is meant by “of their burdens and histological analysis”? 

Line 90 – “ The application of other methods such as immunohistochemistry (IHC), flow cytometry, and biochemical and molecular assays, either sometimes or additionally used, are questionable[ [11].”   - What do the authors mean by “questionable”?  Do the authors mean “less reliable” at detecting micro or macro metastases or something similar?  Are these the “other quantitative methods” the authors talk about in the previous sentence?

Lines 115-117 – “The p63” and “The p63 is a transcription factor…”  Remove the word “The” before p63.  The “the” was only needed when the word “gene” came after p63 (the p63 gene). 

Line 161-162 – “….and vascular endothelial growth factor (VEGF), as well as led to downregulation the proenzyme and active form of MMP-2, and decreased expression of the VEGF protein.”  Consider changing to:  “…and vascular endothelial growth factor (VEGF), and led to downregulation of the proenzyme and active form of MMP-2 and decreased expression of the VEGF protein.” 

Line 167 – what is the “CAM assay”?  The authors introduce “chick embryo chorioallantoic membrane (CAM)” as one of the “other hosts occasionally used in metastatic research” on line 78-79, but it is not clear what the actual assay is or what is measures.  A brief mention of this assay in the introduction where the authors introduce other assays (lines 52-54) would be helpful.

Line 183– “….and the human CMV promoter demonstrated decreased motility of knock-down cells….” – What does that mean?  How did the CMV promotor do this?  The Authors removed the phrase in parenthesis “to gain over expression of Snail2”, suggesting that the CMV promotor was part of a plasmid that contained the Snail2 gene and then cells were transfected with the plasmid to gain overexpression of Snail2.  Is that accurate?  If so, the authors should separate these 2 sentences regarding the Snail2 small interfering RNA and the Snail2 overexpression plasmid. 

Consider changing lines 182-188:   “Downregulation of Snail2 by small interfering RNA demonstrated decreased motility of D17 cells in the scratch assay, suggesting that Snail2 is required for migration [26 – or is this 27?].  Overexpression of Snail2 using a Snail2 plasmid with CMV promotor, however, did not increase the motility of D17 cells.  This could be due to the fact that D17 cells metastasize to the lungs and their migration rate cannot be enhanced by increasing of Snail2 activity, as they may already be migrating at their highest rate [26].”   Is #26 the correct reference here?  Is this the same study as the one evaluating Snail2 in canine OSA with IHC?  Or should this actually be ref #27?

Line 200 – remove “matrix”

Line 201 – The section regarding MMPs seems to be missing a title and was merged into the Snail2 topic paragraph.  Was that intentional?

Line 207 – “…in all 30 high grade…” – change to “…in 30 high grade…”.  The “all” doesn't make sense without previous context. 

Line 275-276 – Are those canine OSA cells?

Line 279-282 – The sentence is confusing.  Consider changing to “ Data regarding IGFR-1 in canine OSA is limited to one study where immunostaining was performed and showed a shorter survival in canine appendicular OSA patients with high IGFR-1 immunoexpression compared to those with low IGFR-1 [18]. Unfortunately, this study lacked any further information….”

Line 365 “…phenotype in canine OSA, but a potential…”  Change “but” to “although”

Line 384 – change to “non-invasive canine OSA cells”.

Line 388 – remove “matrix”

Line 394-396 – “Stimulating the production of MMPs in neoplasias, basigin implicates tumor invasiveness and the promotion of metastasis”  - what is “basigin”?  Do the authors mean “benign neoplasias”?  Then do the authors also mean “increases”, not “implicates”?

Line 396 – sentence beginning with “Roy et al [34] established” – This is a very long sentence and it grammatically falls apart toward the end.  Please separate into 2 or more sentences.

Line 413 – remove “matrix”

Line 416-420 – “Furthermore, the potential role of collagen in cell migration was further confirmed in a WHA, in which the HMPOS cell line treated with aP4AH1-protein inhibitor involved collagen biosynthesis.”   This is an awkward sentence that needs grammar correction. Consider changing to “The potential role of collagen in cell migration was further confirmed by experiments treating canine OSA cells with an inhibitor of P4AH1, which is a protein involved in collagen biosynthesis [76]. Inhibitor-treated HMPOS…..”   The citation on line 420 can be moved to line 418 as in the suggested change above. 

Line 434-438 – Suggested Change – “Jackson et al. [66] observed a correlation between higher ezrin mRNA expression and invasive phenotype and found that inducing ezrin expression in non-invasive canine OSA cells enhanced their invasive potential.”

Line 438 – the sentence starting with “Ezrin phosphorylation is regulated…” should be a new paragraph.

Line 445-447 -  “…was performed on four patient derived canine OSA cell lines with confirmed expression of ezrin, treated with specific small molecule inhibitor of PKC…”   This is an awkward sentence, consider changing to “…was performed on four patient derived, ezrin expressing canine OSA cell lines treated with specific small molecule inhibitor of PKC…”   

Paragraph starting at line 451  - An introductory sentence regarding p-ERM would be helpful here, rather than jumping into data.  The data referenced in #82 here is confusing.  Was the increased ezrin and p-ERM levels observed in the mouse lung tissue or the HMPOS tumor tissue that metastasized to the lungs?  It looks like the Ezrin and p-ERM levels increased at 1 week post IT (intra-tibial?) implantation, then decreased at 2 and 4 weeks post implantation, is that correct?  Did the levels return to normal levels; were they similar to pre-implantation levels at 4 weeks?  The first few sentences of this paragraph should be re-written for clarification.

Paragraph starting at line 466 – this paragraph can be merged with the one above it as they are related topics. 

Line 468 – “… in 83% of canine OSA samples (from 73 dogs with spontaneous….”  Can be changed to “… in 83% (XX/73) of spontaneous primary canine OSA samples.”  The “XX” being the number of positive cases in that study.

For reference #80 – the authors acknowledge that the Ezrin and P-ERM expression data was limited to primary OSA, and not metastatic OSA.  However, does that study reveal whether or not those canine patients also had metastatic disease (although this likely can be assumed).  I mention this because, combined with the study data of the previous paragraph regarding Ezrin and P-ERM levels increasing during the 1st week after tumor transplantation, the observed increase in ezrin in the primary tumor may further support the idea that Ezrin is involved early in the metastatic process, possibly something that allows primary tumor cells to intravasate and transport to the lungs, after which ezrin levels decrease as the metastatic tumor is established.   Another reason why merging that last paragraph with the one above it makes sense.

Line 485 – “ OSA Cell line’s”  - remove apostrophe for “lines”, but add a comma after lines – “In metastatic canine OSA cell lines, TAZ depletion…”

Line 487-488  - “However, the effect of TAZ depletion on cell migration was not observed in canine OSA cell lines of primary tumor origin [87]” – Are the authors referring to fresh cultured (“primary”) tumor samples here, as opposed to established cell lines, or do the authors mean the effect of TAZ depletion on cell migration was only observed in canine OSA cells of metastatic origin, but not primary (non-metastatic) origin?  Please clarify. 

Line 505 – “…determined the expression of TrKA in 10 of 15….”  Do the authors mean “determined high expression”?   

Line 516 – remove comma after “assessed”

Line 517 – what is “directional migration”, as opposed to migration in general?

Line 519 – spelling “folllowing”

Line 539-541 –  Are those canine or human cells that overexpressed miR-9 and exhibited enhanced invasiveness and migration?  The second half of that sentence regarding OSA8 can be separated into its own sentence.  I don't see how the second part “further confirms” the second part and so that phrase is out of place here.  Together, the 2 results “confirm” a role for miR-9 in migration and invasion, which is stated in the next sentence.

Line 547 – The sentence starting with “Presumably….” is awkward – “presumably” is not the right word to use as it implies there is no data to support the statement.   I would suggest changing to “One protein regulated by miR-9 that may promote the metastatic phenotype is gelsolin.  Diminishment of gelsolin in several….” 

The entire miRNA section (lines 524-563) jumps around between miR-9, miR-34a, and gelsolin, making it difficult to follow.  I suggest separating into 2 paragraphs – one paragaph that includes the miR-9 and gelsolin data, and a second that covers the miR-34a data.

Line 592 – add “is” - “….Stokal and collaborators imply is indispensable for further investigation.”

Line 668 – Suggest leaving in “matrix metalloproteinase” here (don't use “MMP”) because you are describing an abbreviation.  Best not to use another abbreviation to describe an abbreviation.

Author Response

Dear Reviewer,

Thank you very much for your hard work in the review process and performing such a detailed review, which improved the final version of the manuscript. We addressed each of your comments below:

Line Comments:

Comment 1 : Figure 1 – The author’s mention that this figure was removed, but the figure appears to still be present in the re-write. 

Answer 1: In our revised version of the manuscript the Figure 1 was delayed. However, the Editor asked to improve the Figure 1 and include it, as a result in the final version we added an improved version of Figure 1 by making changes based on previous comments from both Reviewers (Figure 2 in the final version of the manuscript).

Comment 2: Line 65-70 – “Moreover, the tumor implantation in both canine and human OSA is advantageous over the commonly used injection of OSA cell suspensions into the femur or tibia of mice…”  The way it is written, the authors suggest that injecting humans and dogs with tumor cells is advantageous to injecting mice with OSA cells.  The word “implantation” suggests tumors are being injected into humans and dogs.    Do the authors mean “…the spontaneous tumor growth of both canine and human OSA is advantageous….”?  Or are the authors trying to describe a different method of implanting human and canine OSA cells into mice that differs from the method of injecting them into the femur or tibia.  It is not clear what the authors are trying to say here.

Answer 2: We clarify the information in the sentence leaving only the most important information in context of the article: “Moreover, the primary canine or human OSA tumor implantation into the side of an orthotopic mouse model exemplifies all of the metastatic cascade steps[16].” Indeed, by implantation we meant implantation of the tumor fragment.

Comment 3: Line 78 – Remove the “:”

Answer 3: We removed “the” (line 78 in the new version of the manuscript)

Comment 4: Line 87 – “Monitoring of in vivo macrometastases is performed with non-invasive imaging methods such as MRI or bioluminescence.”   - Do the authors have a reference for this statement?  X-rays are standard protocol for monitoring lung metastasis in dogs, not MRI or bioluminescence. 

 Answer 4: The MRI or bioluminescence were the methods to monitor macrometastasis in preclinical models, we added the information that it is for preclinical models (lines 83-85 of the final version of the manuscript), not for dogs, and we added the citation [11]. Of course, we totally agree with the Reviewer that X-Ray is a standard technique for monitoring lung metastasis in dogs. We added this information to the manuscript (lines 82-83 in the final version of the manuscript) and we provided Fig. 1C with the X-Ray of canine OSA metastasis to the lungs.

Comment 5: Line 88 – “Both macro and micrometastases can be detected by other quantitative measurements of their burdens and histological analysis” – What other quantitative measurements are the authors referring to here?  Reference?  What is meant by “of their burdens and histological analysis”? 

Answer 5: Indeed, we agree with the Reviewer that the meaning of the sentence was unclear. We rewrote the sentence to clarify the information provided (lines 87-89 in the final version of the manuscript). We added the citation [11].

Comment 6: Line 90 – “ The application of other methods such as immunohistochemistry (IHC), flow cytometry, and biochemical and molecular assays, either sometimes or additionally used, are questionable[ [11].”   - What do the authors mean by “questionable”?  Do the authors mean “less reliable” at detecting micro or macro metastases or something similar?  Are these the “other quantitative methods” the authors talk about in the previous sentence?

Answer 6:  By questionable we meant questionable predictive power (debatable sensitivity and specificity)- citation 11. Indeed, those were the other quantitative methods that we spoke of in the previous sentence. However, as it was difficult to read, so  we rewrote the sentences to clarify it (lines 89-92 of the final version of the manuscript).

Comment 7: Lines 115-117 – “The p63” and “The p63 is a transcription factor…”  Remove the word “The” before p63.  The “the” was only needed when the word “gene” came after p63 (the p63 gene). 

Answer 7: we removed the “the” beforep63 (line 116-117 in the final version of the manuscript)

Comment 8: Line 161-162 – “….and vascular endothelial growth factor (VEGF), as well as led to downregulation the proenzyme and active form of MMP-2, and decreased expression of the VEGF protein.”  Consider changing to:  “…and vascular endothelial growth factor (VEGF), and led to downregulation of the proenzyme and active form of MMP-2 and decreased expression of the VEGF protein.” 

Answer 8: we changed it according to the Reviewers’ suggestion (lines 152-153 in the final version of the manuscript)

Comment 9: Line 167 – what is the “CAM assay”?  The authors introduce “chick embryo chorioallantoic membrane (CAM)” as one of the “other hosts occasionally used in metastatic research” on line 78-79, but it is not clear what the actual assay is or what is measures.  A brief mention of this assay in the introduction where the authors introduce other assays (lines 52-54) would be helpful.

Answer 9: We added a brief introduction about the use of the CAM assay in anti-metastatic studies. (lines 70-74 in the final version of the manuscript)

Comment 10: Line 183– “….and the human CMV promoter demonstrated decreased motility of knock-down cells….” – What does that mean?  How did the CMV promotor do this?  The Authors removed the phrase in parenthesis “to gain over expression of Snail2”, suggesting that the CMV promotor was part of a plasmid that contained the Snail2 gene and then cells were transfected with the plasmid to gain overexpression of Snail2.  Is that accurate?  If so, the authors should separate these 2 sentences regarding the Snail2 small interfering RNA and the Snail2 overexpression plasmid. 

Consider changing lines 182-188:   “   Is #26 the correct reference here?  Is this the same study as the one evaluating Snail2 in canine OSA with IHC?  Or should this actually be ref #27?

Answer 10:  We rewrote the sentence as suggested by the Reviewer (lines 169-174 in the final version of the manuscript) and corrected the citations (line 179 and 182 in the final version of the manuscript)– indeed it should be [27](in the new version of the manuscript its [28] as we added one more citation of Kim et al when describing the CAM model), not [26]

Comment 11: Line 200 – remove “matrix”

Answer 11: We removed the word “matrix” as indicated (line 191, 198 in the final version of the manuscript)

Comment 12: Line 201 – The section regarding MMPs seems to be missing a title and was merged into the Snail2 topic paragraph.  Was that intentional?

Answer 12:  Yes, we did it in response to a comment from the other Reviewer, who said that in his opinion there is lack of studies on the real role of MMPs in canine metastasis cascade, and the information on MMPs should be merged with another section.

Comment 13: Line 207 – “…in all 30 high grade…” – change to “…in 30 high grade…”.  The “all” doesn't make sense without previous context. 

Answer 13: We removed word “all” as indicated (line 198 in the final version of the manuscript)

Comment 14: Line 275-276 – Are those canine OSA cells?

Answer 14: Yes, we added the information that those are canine OSA cell lines (line 248, 250 in the final version of the manuscript)

Comment 15: Line 279-282 – The sentence is confusing.  Consider changing to “ Data regarding IGFR-1 in canine OSA is limited to one study where immunostaining was performed and showed a shorter survival in canine appendicular OSA patients with high IGFR-1 immunoexpression compared to those with low IGFR-1 [18]. Unfortunately, this study lacked any further information….”

Answer 15: We changed the sentence as suggested by the Reviewer (lines 253-256 in the final version of the manuscript)

Comment 16: Line 365 “…phenotype in canine OSA, but a potential…”  Change “but” to “although”

Answer 16: We changed “but” to “although” as suggested by the Reviewer (line 310 in the final version of the manuscript)

Comment 17: Line 384 – change to “non-invasive canine OSA cells”.

Answer: We added word “canine” making it “non-invasive canine OSA cells” as indicated by the Reviewer (line 325 in the final version of the manuscript)

Comment 18: Line 388 – remove “matrix”

Answer: we removed word “matrix” as indicated by the Reviewer (line 329 in the final version of the manuscript)

Comment 19: Line 394-396 – “Stimulating the production of MMPs in neoplasias, basigin implicates tumor invasiveness and the promotion of metastasis”  - what is “basigin”?  Do the authors mean “benign neoplasias”?  Then do the authors also mean “increases”, not “implicates”?

Answer: Basigin is the other name of CD147, which is previously mentioned previously in the manuscript (line 329 in the final version of the manuscript). However, as it may be confusing, so we changed the name “basigin” into “CD147” (line 335 in the final version of the manuscript).

Comment 20: Line 396 – sentence beginning with “Roy et al [34] established” – This is a very long sentence and it grammatically falls apart toward the end.  Please separate into 2 or more sentences.

Answer 20: We divided the sentence beginning with Roy et al. [34] into 2 sentences as indicated by the Reviewer (lines 336-342).

Comment 20: Line 413 – remove “matrix”

Answer 20: We removed word “matrix” as indicated by the Reviewer (line 351 of the final version of the manuscript)

Comment 21: Line 416-420 – “Furthermore, the potential role of collagen in cell migration was further confirmed in a WHA, in which the HMPOS cell line treated with aP4AH1-protein inhibitor involved collagen biosynthesis.”   This is an awkward sentence that needs grammar correction. Consider changing to “The potential role of collagen in cell migration was further confirmed by experiments treating canine OSA cells with an inhibitor of P4AH1, which is a protein involved in collagen biosynthesis [76]. Inhibitor-treated HMPOS…..”   The citation on line 420 can be moved to line 418 as in the suggested change above. 

Answer 21:We rewrote the sentence (lines 354-356 of the final version of the manuscript) and moved the citation [76] as indicated by the Reviewer (from line 359 into 357 of the final version of the manuscript) 

Comment 22: Line 434-438 – Suggested Change – “Jackson et al. [66] observed a correlation between higher ezrin mRNA expression and invasive phenotype and found that inducing ezrin expression in non-invasive canine OSA cells enhanced their invasive potential.”

Answer 22: We change the sentence as indicated by the Reviewer (lines 372-374 of the final version of the manuscript)

Comment 23: Line 438 – the sentence starting with “Ezrin phosphorylation is regulated…” should be a new paragraph.

Answer 23: We made a new paragraph starting with the sentence “Ezrin phosphorylation is regulated…” (line 382 of the final version of the manuscript)

Comment 24: Line 445-447 -  “…was performed on four patient derived canine OSA cell lines with confirmed expression of ezrin, treated with specific small molecule inhibitor of PKC…”   This is an awkward sentence, consider changing to “…was performed on four patient derived, ezrin expressing canine OSA cell lines treated with specific small molecule inhibitor of PKC…”   

Answer:We rewrote the sentence as indicated by the Reviewer (line 384-386 in the final version of the manuscript)

Comment 25: Paragraph starting at line 451  - An introductory sentence regarding p-ERM would be helpful here, rather than jumping into data.  The data referenced in #82 here is confusing.  Was the increased ezrin and p-ERM levels observed in the mouse lung tissue or the HMPOS tumor tissue that metastasized to the lungs?  It looks like the Ezrin and p-ERM levels increased at 1 week post IT (intra-tibial?) implantation, then decreased at 2 and 4 weeks post implantation, is that correct?  Did the levels return to normal levels; were they similar to pre-implantation levels at 4 weeks?  The first few sentences of this paragraph should be re-written for clarification.

Answer 25: We described the role of the phosphorylation of ezrin both in the previous and this paragraph (lines 377-383 of the final version of the manuscript). We rewrote the sentences about ezrin and p-ERM levels to clarify that they increased in the HMPOS tumor tissue that metastasize to the lungs of mice (lines 389-393 in the final version of the manuscript). Indeed, the ezrin and p-ERM levels increased at 1 week post transplantation and then only p-ERM level decreased after 2 and 4 weeks. We also rewrote these sentences to clarify it (lines 392-394 in the final version of the manuscript).

Comment 26: Paragraph starting at line 466 – this paragraph can be merged with the one above it as they are related topics. 

Answer 26: We merged the paragraphs as indicated by the Reviewer (line 397 of the final version of the manuscript)

Comment 27: Line 468 – “… in 83% of canine OSA samples (from 73 dogs with spontaneous….”  Can be changed to “… in 83% (XX/73) of spontaneous primary canine OSA samples.”  The “XX” being the number of positive cases in that study.

 Answer 27: We rewrote the sentence as indicated in the study (lines 397-399 in the final version of the manuscript). However, we didn’t add the exact number of dogs with positive cases as the Authors of the article do not provide exact number, so when calculating how they wrote that should be

Comment 28: For reference #80 – the authors acknowledge that the Ezrin and P-ERM expression data was limited to primary OSA, and not metastatic OSA.  However, does that study reveal whether or not those canine patients also had metastatic disease (although this likely can be assumed).  I mention this because, combined with the study data of the previous paragraph regarding Ezrin and P-ERM levels increasing during the 1st week after tumor transplantation, the observed increase in ezrin in the primary tumor may further support the idea that Ezrin is involved early in the metastatic process, possibly something that allows primary tumor cells to intravasate and transport to the lungs, after which ezrin levels decrease as the metastatic tumor is established.   Another reason why merging that last paragraph with the one above it makes sense.

Answer 28: Dogs included in the study had no evidence of pulmonary metastasis on thoracic radiographs and no clinical evidence on metastasis to the other sides when entering the study. However, as the Reviewer suggested, it is very possible that the animals had micrometastasis that couldn’t be detected clinically and that those findings on ezrin and p-ERM expression could further support the idea that ezrin is involved in the early metastatic process. We added these suggestions to the manuscript (lines 400-410 in the final version of the manuscript).

Comment 29: Line 485 – “ OSA Cell line’s”  - remove apostrophe for “lines”, but add a comma after lines – “In metastatic canine OSA cell lines, TAZ depletion…”

Answer 29: We corrected the sentence as indicated by the Reviewer (line 421 in the final version of the manuscript)

Comment 29: Line 487-488  - “However, the effect of TAZ depletion on cell migration was not observed in primary tumor-derived OSA cell lines [87]” – Are the authors referring to fresh cultured (“primary”) tumor samples here, as opposed to established cell lines, or do the authors mean the effect of TAZ depletion on cell migration was only observed in canine OSA cells of metastatic origin, but not primary (non-metastatic) origin?  Please clarify. 

Answer 29: We clarified the sentence (lines 420-424 in the final version of the manuscript). Indeed, we meant canine OSA cell lines derived from primary tumors (non-metastatic).

Comment 30: Line 505 – “…determined the expression of TrKA in 10 of 15….”  Do the authors mean “determined high expression”?   

Answer 30: We cited this statement after the Authors of the study, but to be precise they did IHC staining, so we clarified the sentence changing expression into “positive membranous immunostaining of most of OSA cells”. We also added % of positively stained samples for both OSA cell lines (66%) and primary OSA (75%) (lines 435-439 in the final version of the manuscript).

Comment 31: Line 516 – remove comma after “assessed”

Answer 31: The comma was removed (line 445 in the final version of the manuscript)

Comment 32: Line 517 – what is “directional migration”, as opposed to migration in general?

 Answer 32:”Conceptually, directional cell migration has two sources: intrinsic cell directionality of migration and external regulation. Intrinsic directionality is observed when cells respond to a non-directional motogenic signal 3, such as the uniform application of platelet-derived growth factor (PDGF) 4, that triggers the basic motility machinery in the absence of any external guiding factor. Random migration occurs when a cell possesses relatively low intrinsic directionality.”(Petrie et al., 2009). There are also different methods used to assess migration in general and directional migration, that’s why we precise within the article when the directional, not random migration, was assessed.

Petrie RJ, Doyle AD, Yamada KM. Random versus directionally persistent cell migration. Nat Rev Mol Cell Biol. 2009;10(8):538-549. doi:10.1038/nrm2729

Comment 33: Line 519 – spelling “folllowing”

Answer 33: We corrected the spelling of the word “following” (line 454 in the final version of the manuscript)

Comment 34: Line 539-541 –  Are those canine or human cells that overexpressed miR-9 and exhibited enhanced invasiveness and migration?  The second half of that sentence regarding OSA8 can be separated into its own sentence.  I don't see how the second part “further confirms” the second part and so that phrase is out of place here.  Together, the 2 results “confirm” a role for miR-9 in migration and invasion, which is stated in the next sentence.

Answer: We added word “canine” before cell lines to clarify that those are canine cell lines (line 462 in the final version of the manuscript). We removed the phrase “further confirms” and rewrote the sentence (lines 508-511 of the final version of the manuscript)

Comment 35: Line 547 – The sentence starting with “Presumably….” is awkward – “presumably” is not the right word to use as it implies there is no data to support the statement.   I would suggest changing to “One protein regulated by miR-9 that may promote the metastatic phenotype is gelsolin.  Diminishment of gelsolin in several….” 

Answer 35: We changed the sentence into the one suggested by the Reviewer (line 465- 472 of the final version of the manuscript) 

Comment 36: The entire miRNA section (lines 524-563) jumps around between miR-9, miR-34a, and gelsolin, making it difficult to follow.  I suggest separating into 2 paragraphs – one paragraph that includes the miR-9 and gelsolin data, and a second that covers the miR-34a data.

Answer 36: We separated the miRNA section into 3 paragraphs: Introducing the role of miRNA (1), miR-9 and gelsin  (2) and miR-34a (3) to make it easier to follow (lines 453-484 in the final version of the manuscript) 

Comment 37: Line 592 – add “is” - “….Stokal and collaborators imply is indispensable for further investigation.”

Answer 37: We added the word “is” as indicated by the Reviewer (line 511 in the final version of the manuscript) 

Line 668 – Suggest leaving in “matrix metalloproteinase” here (don't use “MMP”) because you are describing an abbreviation.  Best not to use another abbreviation to describe an abbreviation.

Answer 38: We changed MMP into matrix metalloproteinase as suggested by the Reviewer (line 550 of the final version of the manuscript)